# Pathogenic mutations of human phosphorylation sites affect protein–protein interactions

Trendelina Rrustemi[1], Katrina Meyer [1,5], Yvette Roske[1], Bora Uyar [1], Altuna Akalin [1], Koshi Imami [2,6], Yasushi Ishihama [2], Oliver Daumke [1,3] & Matthias Selbach [1,4] ✉

Despite their lack of a defined 3D structure, intrinsically disordered regions (IDRs) of proteins play important biological roles. Many IDRs contain short linear motifs (SLiMs) that mediate protein-protein interactions (PPIs), which can be regulated by post-translational modifications like phosphorylation. 20% of pathogenic missense mutations are found in IDRs, and understanding how such mutations affect PPIs is essential for unraveling disease mechanisms. Here, we employ peptide-based interaction proteomics to investigate 36 disease-associated mutations affecting phosphorylation sites. Our results unveil significant differences in interactomes between phosphorylated and non-phosphorylated peptides, often due to disrupted phosphorylation-dependent SLiMs. We focused on a mutation of a serine phosphorylation site in the transcription factor GATAD1, which causes dilated cardiomyopathy. We find that this phosphorylation site mediates interaction with 14-3-3 family proteins. Follow-up experiments reveal the structural basis of this interaction and suggest that 14-3-3 binding affects GATAD1 nucleocytoplasmic transport by masking a nuclear localisation signal. Our results demonstrate that pathogenic mutations of human phosphorylation sites can significantly impact protein-protein interactions, offering insights into potential molecular mechanisms underlying pathogenesis.

Understanding protein function in health and disease is a key challenge in the post-genomic era[1]. Omics techniques provide a wealth of data, but mechanistic understanding is often lagging behind. For example, although sequencing technologies have identified numerous single amino acid variants (SAVs), their functional implications remain mostly unknown, even when they have been linked to disease[2–4]. Proteins are also modified by posttranslational modifications (PTMs) such as phosphorylation, and proteomics can now routinely identify tens of thousands of phosphorylation sites[5–8]. However, most sites have no known kinase or biological function[9]. Hence, while genomic and proteomic technologies provide abundant information about SAVs and PTMs, respectively, how these changes affect protein function remains largely unexplored.

The classical sequence-structure-function paradigm posits that amino acid sequences determine protein structure and therefore protein function. Accordingly, the impact of SAVs and PTMs on protein function is often investigated from a structural angle. However, around 40% of the proteome consists of intrinsically disordered regions (IDRs)

[1]Max Delbrück Center (MDC), Robert-Rössle-Str. 10, 13125 Berlin, Germany. [2]Graduate School of Pharmaceutical Sciences, Kyoto University, Kyoto 606-8501, Japan. [3]Freie Universität Berlin, Institute of Chemistry and Biochemistry, Takustraße 6, Berlin, Germany. [4]Charité-Universitätsmedizin Berlin, 10117 Berlin, Germany. [5]Present address: Max Planck Institute for Molecular Genetics, Ihnestraße 63, 14195 Berlin, Germany. [6]Present address: RIKEN Center for Integrative Medical Sciences, Yokohama 230-0045 Kanagawa, Japan. ✉e-mail: matthias.selbach@mdc-berlin.de

that have low sequence complexity[10], and over 20% of known disease-associated and cancer driver mutations affect amino acid residues in IDRs[11,12]. Amino acids in IDRs are also frequently modified by PTMs[13–15]. Due to the lacking structure-function relationship, understanding how SAVs and PTMs in IDRs affect protein function is especially challenging.

It is now well established that IDRs are critically involved in virtually every cellular process[16]. They achieve this in a number of different ways such as the induction of structural changes in adjacent structured regions, transitioning from disorder-to-order, and/or by mediating protein-protein interactions[17–19]. Protein-protein interactions in IDRs are mediated by so-called short linear motifs (SLiMs) – sequence stretches shorter than ten amino acids with simple specificity determinants that are recognized by cognate domains in interacting proteins[20,21]. Importantly, many SLiM-mediated interactions are dynamically regulated by PTMs, and the dynamic interplay between specific PTMs (e.g. tyrosine phosphorylation) and recruitment of protein readers with cognate domains (e.g. SH2 domains) plays a pivotal role in cell signaling[22]. In fact, more than 20% of validated SLiMs in the Eukaryotic Linear Motif (ELM) database are post-translationally modified, with phosphorylation being the most common modification type[23].

A number of different experimental approaches to map the SLiM-based interactome have been established[21]. One such approach is peptide-based interaction proteomics (reviewed by[24]). It employs synthetic peptides corresponding to IDRs of interest that are used to pull-down interacting proteins from complex protein lysates. In combination with the parallel synthesis of peptides on cellulose membranes (SPOT synthesis)[25], the throughput of peptide-based interaction proteomics can be greatly increased[24,26,27]. This Protein Interaction Screen on Peptide Matrix (PRISMA) method has been applied in a number of recent studies[28–30]. For example, we used PRISMA to study how pathogenic mutations in IDRs affect protein-protein interactions (PPIs). This revealed that a pathogenic point mutation in the glucose transporter GLUT1 causes GLUT1 deficiency syndrome by creating a SLiM that recruits adaptor proteins and mediates GLUT1 endocytosis[30]. A key advantage of peptide pulldown approaches is that the peptides can be synthesized in modified forms, enabling direct assessment of the impact of PTMs on PPIs[27,28,31–34].

Disease-associated SAVs in IDRs are enriched at interaction interfaces, supporting the view that many mutations in IDRs cause disease by affecting PPIs[35]. However, if and how these perturbed interactions also involve PTMs is not well understood. Since both disease-associated SAVs and PTMs can affect PPIs, we reasoned that studying SAVs that affect known phosphorylation would be particularly interesting.

In this work, we investigate the interplay of disease-associated SAVs and PTMs on protein-protein interactions. Specifically, we select SAVs affecting known phosphorylation sites and ask if the SAV and/or the phosphorylation state of the site changes protein-protein interactions. To this end, we use PRISMA to directly compare the interactome of the wild-type, mutated and phosphorylated site.

## Results

### PRISMA for phosphorylation and disease SAV interactions

To assess how disease-related SAVs of protein phosphorylation sites affect protein-protein interactions, we first selected pathogenic missense mutations of known serine, threonine or tyrosine phosphorylation sites. To this end, we used the PTMvar dataset from PhosphositePlus that maps posttranslational modification sites to disease-associated genetic variants[36], see Methods for more details on variant classification. At the time, this dataset included 33,359 entries, including 12,658 disease-associated mutations from various databases (COSMIC, Uniprot Humsavar, TCGA, cBio) and seven types of posttranslational modifications. After filtering for mutations in intrinsically

disordered regions, we obtained 1965 mutations, of which 126 directly affected the phosphorylated amino acid. We then selected mutations that had no other annotated modifications within seven amino acids of the target residue. This filtering resulted in a final set of 38 disease candidates derived from 34 different proteins (Fig. 1A, Supplementary Data 1).

To experimentally investigate how mutation and/or phosphorylation of these sites affects protein-protein interactions, we employed a peptide-based interaction screen[24]. In particular, we adapted the Protein interaction screen on peptide matrix (PRISMA) set-up, where peptides are synthesized on a cellulose matrix that is used to pull-down interaction partners from protein extracts directly[27,30,37]. To evaluate the impact of both the disease-associated mutation and phosphorylation, we designed an experiment that allowed us to compare interactions of all three peptide states (wild-type non-phosphorylated, wild-type phosphorylated, and mutated) directly with each other (Fig. 1C). All 38 peptides were synthesized on cellulose membranes via SPOT synthesis[25] as 15-mers with the phosphorylation site in the central position.

We also included the three forms of a well-characterized EGFR-derived phosphopeptide as a positive control[31]. Thus, the cellulose membranes contained $3 \times 39 = 117$ different peptide spots. For quantification, we employed stable isotope labeling with amino acids in cell culture (SILAC)[38]. We used lysates of unlabelled (light or L), medium heavy (M), or heavy (H) SILAC-labeled HEK-293 cells. Each of the three differently SILAC-labeled cell lysates was incubated with a different copy of the cellulose membrane to pull-down specific interaction partners. After washing, peptide spots with their bound proteins were excised and combined with the two other peptide states from the other two membranes. We always combined the pull-downs of the wild-type non-phosphorylated, wild-type phosphorylated, and mutated peptides from three membranes into one sample. In this way, SILAC-based quantification allows us to directly evaluate differences in binding partners across the three peptide states.

We analyzed all 117 combined samples by high-resolution shotgun proteomics. Two of the 38 mutations were mixed up and were therefore excluded. In each of the remaining 111 samples, we identified between 300 and 1000 proteins (Supplementary Fig. 1A), which results in a total number of about 70,000 putative protein-peptide interactions. The correlation of label-free quantification (LFQ) values between replicates was considerably higher than the correlation between different peptide pulldowns, demonstrating good reproducibility (Supplementary Fig. 1B). To further validate the data we grouped the ~70,000 putative protein-peptide interactions into two categories: those that can be explained by a matching SLiM-domain pair and those that cannot. More specifically, we considered all known Eukaryotic Linear Motif (ELM) classes (356 classes) and all known PFAM domains (180 unique PFAM domains) that are known to recognize SLiMs from the ELM database[39]. For each pulldown, we then asked if proteins contain a PFAM domain that matches to a SLiM in the respective peptide, i.e. that the interaction can be explained by a SLiM-domain pair (see Supplementary Data 3). We observed higher protein LFQ values, i.e. enrichment, in pulldowns when the peptide exhibited a SLiM matching a protein domain, thus demonstrating that our data provides valuable insights into SLiM-dependent interactions (Supplementary Fig. 1C). To provide a more detailed picture, we looked at the four most frequently observed SLiMs mediating interactions (motif class type "DOC" and "LIG" in ELM) (Supplementary Fig. 2). For example, we observe that proteins containing a WW domain preferentially bind to peptides harboring corresponding motifs. Importantly, 14-3-3, SH2 and WW domain-containing proteins bind preferentially to the phosphorylated form of the peptide, consistent with the phosphorylation-dependent nature of these interactions. In contrast, SH3 domain-containing proteins interact with cognate SLiMs in a phosphorylation-independent manner (Supplementary Fig. 2).

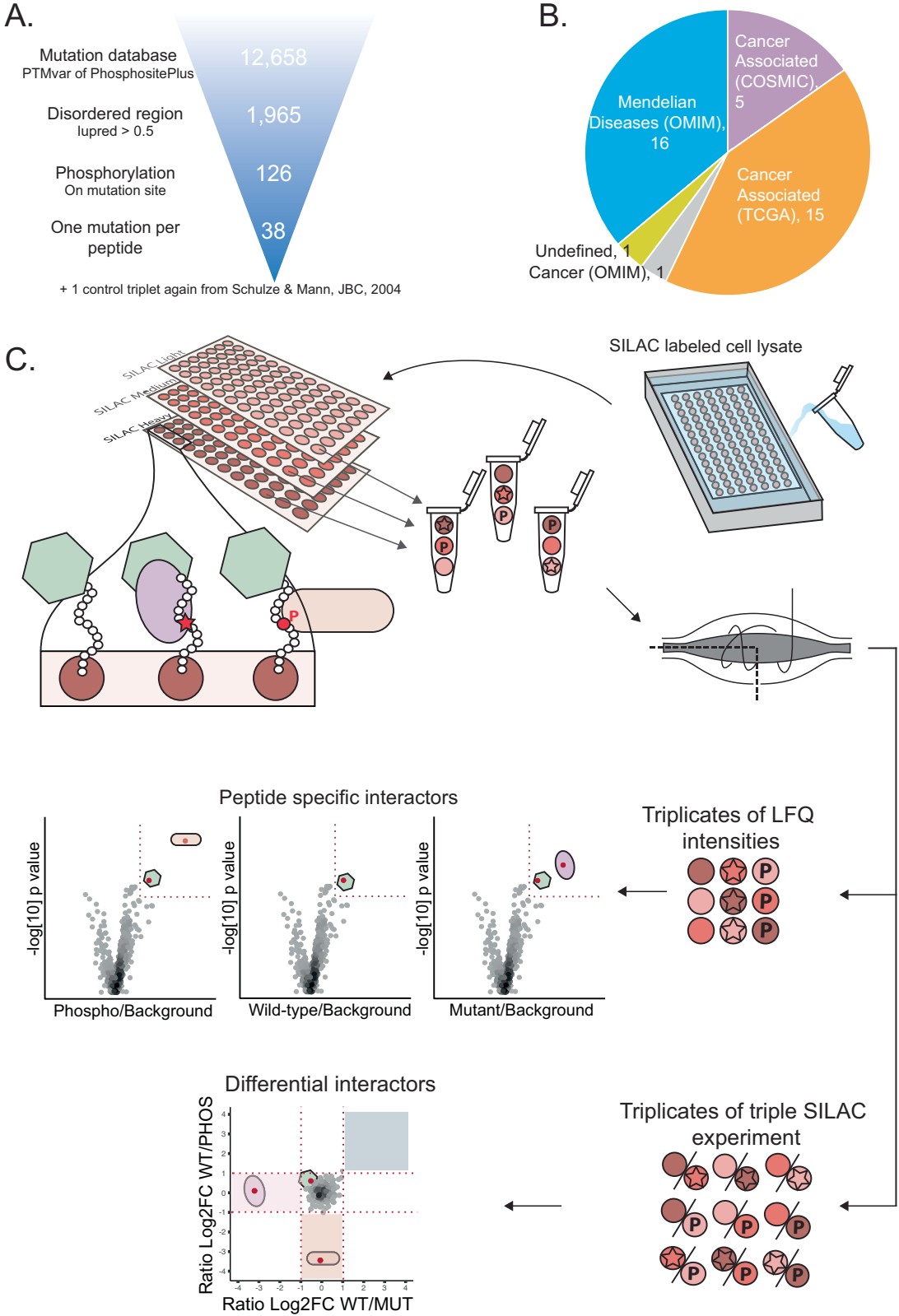

**Fig. 1 | Candidate selection and experimental design. A** Selection scheme for peptide candidates from the PTMvar database of PhosphositePlus. **B** Out of the 38 peptide candidates included in the screen, 20 are associated with cancer, 17 cause Mendelian diseases and for 1 candidate the disease is undefined. **C** A scheme illustrating the experimental design and data analysis of the PRISMA screen. Three peptide states (empty: wild-type non-phosphorylated, P: wild-type phosphorylated, star: mutated).

Overall, these data show that the observed global relationship also holds for individual SLiM-domain pairs.

## Quantification enables detection of specific interactions

Quantification is an efficient means to differentiate specific interaction partners from non-specific contaminants[40–43]. Following our previously published strategy[30], we used two consecutive quantitative filters to identify proteins that exhibit both specific interactions with a particular peptide and are influenced by its phosphorylation and/or mutation state. First, we used label-free quantification (LFQ) to identify proteins that interact specifically with a peptide compared to all other peptides in the screen. To achieve this, we utilized a Wilcoxon test to contrast the proteins derived from a single peptide with the background. For the EGFR control peptide, this LFQ filter identified 2, 2, and 12 specific interaction partners of the wildtype, muted and phosphorylated state, respectively (Fig. 2A). Second, we employed SILAC-based quantification to compare interactions across these three peptide states. We present the SILAC ratios of wildtype versus mutant and wildtype versus phosphorylated states as a scatter plot (Fig. 2B). Differential proteins were identified based on their log2 wt/mut, wt/phos, and phos/mut ratios being either greater than 1, or less than −1. As expected for the tyrosine-phosphorylated EGFR peptide, a number of SH2-domain containing proteins are LFQ-specific interactors of the phosphorylated form (Fig. 2A, right panel), and the SILAC data confirms their phosphorylation-dependent interaction (Fig. 2B). The autophosphorylated EGFR peptide has been shown to bind to the GRB2 protein[31,32,44]. In our analysis, as depicted in the scatter plot, it is evident that GRB2 exhibits the highest phos/wt and phos/mut ratio >20, in all three replicates. MS1 spectra for a GRB2-derived peptide across label swaps are shown as an example (Fig. 2C). In addition to GRB2, we identified several other SH2 domain-containing proteins as binders, including STAT3 and PLCG1, consistent with previous data[32]. More detailed information on the data analysis pipeline can be found in the section 'Data Analysis' of the Methods.

Having validated our quantitative filters for the positive control, we applied this strategy to the entire dataset (Supplementary Figs. 4–39). LFQ-based filtering reduced the ~70,000 interactions to approximately 500 (Fig. 2E and Supplementary Fig. 3B). 70 of the 111 peptides had at least one specific interaction partner and 31 out of 37 peptide triplets had at least one specific and differential binding partner (Fig. 2D and Supplementary Fig. 3A). Among all LFQ-specific interactors, SILAC-based filtering identified 170 interactions to be differential between the three peptide states. Thus, our filtering approach dramatically reduces the number of interactions. Among the specific and differential interactors, 105 preferentially interact with the phosphorylated peptide, 33 with the non-phosphorylated wild-type, and 32 with the mutated peptide (Fig. 2E, Supplementary Fig. 3C and Supplementary Data 2).

## A network of interactions affected by phosphorylation

We present all 170 specific and differential interactions with the 31 peptides in the form of an interaction network (Fig. 3A). To illustrate which interactions can be explained by protein domains binding to a peptide SLiM, we extracted all annotated SLiMs from the peptide sequences and highlighted the proteins in the network that contain matching domains. For example, the network contains three tyrosine phosphorylated peptides (including the EGFR control) that have SH2-domain binding SLiMs, and we identified six SH2 domain-containing proteins as specific and differential interactors (Fig. 3B). Interactors other than those containing SH2 domains likely bind indirectly to these tyrosine phosphorylated peptides. For instance, the interaction between the EGFR peptide and SOS1 is likely mediated through GRB2[45]. We also observed twelve peptides interacting with PIN1 (Peptidyl-prolyl cis/trans isomerase NIMA-interacting 1) (Fig. 3C). 10 of these peptides contain an [pS/T]P motif, known to interact with the WW

domain of PIN1[46]. 8 out of 10 peptides contain the motif at the phosphorylated site, while 2 others contain it in the adjacent sequence. Finally, we were intrigued by the observation that MMTAG2, ARL6IP4, and PC4 interact with many phosphopeptides (9, 7, and 6, respectively). While these proteins do not contain annotated domains known to mediate phosphorylation-dependent binding, they carry regions with compositional bias for basic amino acids and have high predicted isoelectric points (Fig. 3D). These proteins are therefore positively charged in the neutral pH range we used in the pulldown. Since phosphate groups are negatively charged, the observed phosphorylation-dependent interaction could reflect electrostatic effects. In summary, our interaction network contains a number of phosphorylation-dependent interactions that can be explained by SLiM-domain pairs or other specific protein features. Nevertheless, a majority of the identified interactions are distinct, offering significant potential for elucidating the impact of SAVs and/or phosphorylation on protein function.

## S102 in GATAD1 interacts with 14-3-3 proteins

To further investigate the potential function of distinct interactions, we directed our attention to the binding of multiple members of the 14-3-3 family of proteins to a GATAD1-derived phosphopeptide (Fig. 4A and Supplementary Fig. 11). 14-3-3 family proteins are important regulatory molecules involved in a staggering number of cellular processes that interact with target proteins in a phosphorylation-dependent manner[47–49]. GATAD1 is a transcription factor affecting proliferation and cell cycle via controlling AKT signaling[50]. The GATAD1 S102P mutation we investigated in the screen was described to cause dilated cardiomyopathy (DCM) in an autosomal recessive manner in a consanguineous family[51]. Interestingly, the mutation appears to affect the subcellular localisation of GATAD1 in cardiomyocytes of patients carrying this mutation. This is interesting since 14-3-3 proteins have been shown to regulate the subcellular localisation of their binding partners. Experiments in zebrafish provided additional evidence for the pathogenicity of this mutation[52]. However, neither the pathogenic mechanism nor the function of this GATAD1 phosphorylation site are currently known. Intriguingly, although GATAD1 is expressed in many tissues, the only evidence for phosphorylation of this site comes from murine heart tissue, indicative of a heart-specific function[36,53].

To further characterize the interactome of WT and mutated GATAD1 we used affinity purification-mass spectrometry (AP-MS) using the FLAG tag[54]. To this end, we generated stable inducible Flp-In™-293 cells expressing GATAD1 fused to the promiscuous biotin ligase BirA* and FLAG tag in the N-terminus. We created four different GATAD1 variants: The wild-type protein (WT), the disease-associated S102P variant, a non-phosphorylatable S102A variant, and a phosphomimetic S102D variant. Immunofluorescence confirmed that all the BirA*-FLAG-GATAD1 fusion proteins localize to the nucleus (Supplementary Fig. 40). This observation suggests that the point mutations do not affect the nuclear localization of GATAD1. AP-MS studies identified 47 proteins as specific WT GATAD1 interactors (Fig. 4B and Supplementary Data 4). This included several well-known GATAD1 binders such as the transcriptional regulators KDM5A, RBBP7/4 PHF12, and SIN3B. Together, these proteins form the EMSY complex, which binds to H3K4me3-marked, active promoters[33,55]. More globally, gene ontology enrichment analysis of these 47 proteins revealed their involvement in Sin3 complex and transcriptional corepressor activity[33,55], consistent with the known biology of GATAD1.

Overall, the AP-MS data for all the GATAD1 mutant variants was very similar to the wild-type with the same interaction partners identified in all (Fig. 4B). Moreover, we did not detect 14-3-3 family proteins as significant GATAD1 interactors with any of the variants. One possible explanation for this could be that GATAD1 is not phosphorylated on S102 in HEK cells. Indeed, the interactome of wild-type GATAD1 and

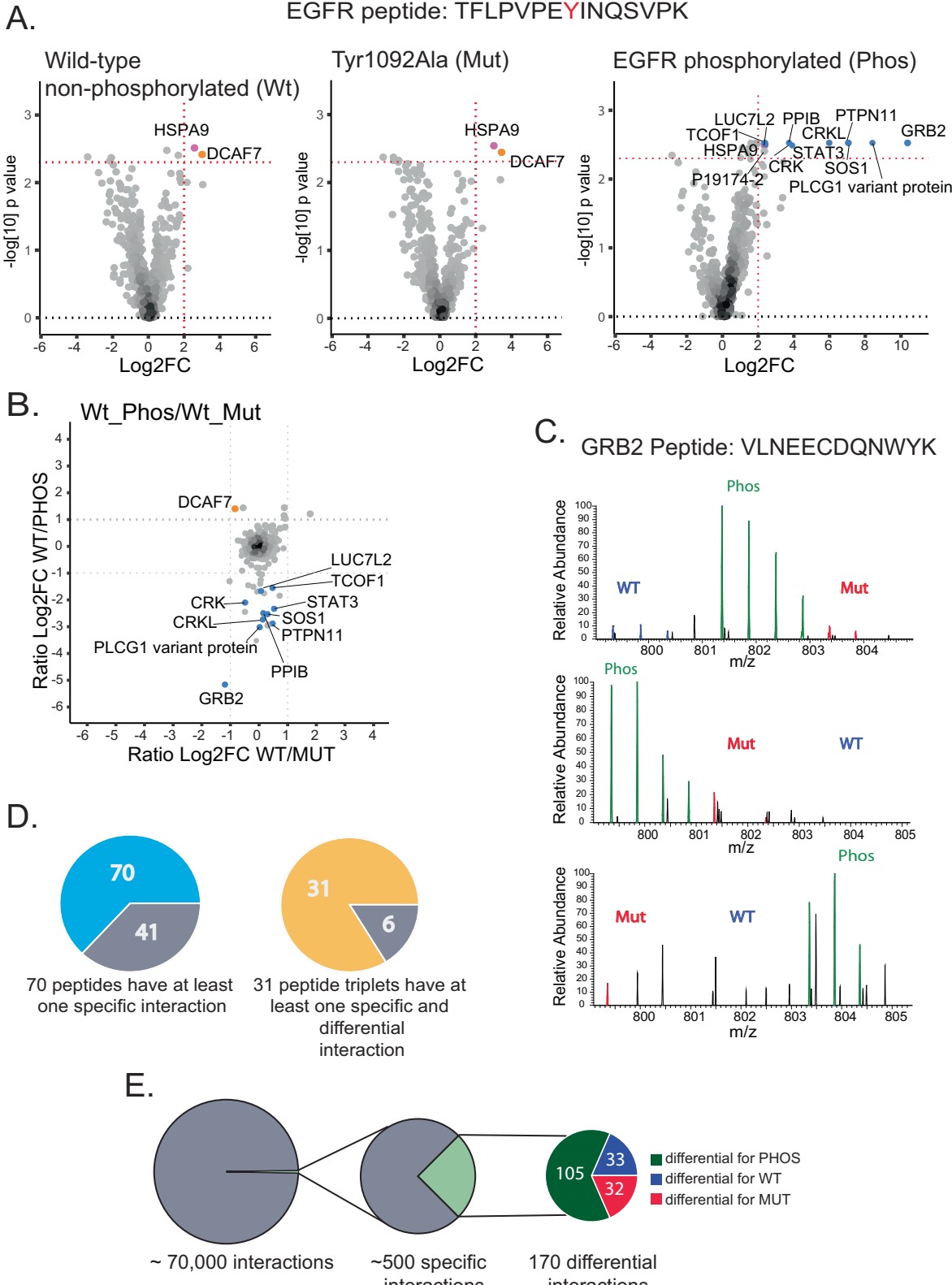

that of the non-phosphorylatable control (S102A) were almost identical (Fig. 4B and Supplementary Data 4). Moreover, even though GATAD1 is widely expressed and multiple phosphorylation sites (T34, T52, S55, S194, S235, Y248) have been identified in different cell lines (HeLa, KG1, K562, MKN-45), the only evidence for S102 phosphorylation comes from murine heart tissue[36,53]. To mimic phosphorylation, we created the S102D mutant that introduces a negative charge and is

thus considered to be phosphomimetic[56,57]. Nevertheless, this particular mutant also did not exhibit any interaction with 14-3-3 proteins. (Fig. 4B).

A complementary approach to assess cellular protein interactions is proximity labelling. In contrast to AP-MS, proximity labeling methods like BioID do not require stable physical interactions. Instead, they capture the 'neighborhood' of proteins in the context of a living cell[58].

**Fig. 2 | Quantification enables the detection of specific interactions. A** Volcano plots, generated after Wilcoxon testing based on LFQ quantification, illustrate that only the phosphorylated form of the EGFR control peptide displays specific interactions with SH2 domain-containing proteins (Log2 fold change >2 and *p*-value < 0.005). Proteins surpassing the LFQ significance threshold are color-coded for clarity (blue: phosphopeptide specific, orange: wild-type and mutated peptide specific, pink: specific for all three peptide forms). **B** Scatter plot depicting SILAC ratios for the positive EGFR control. The *x*-axis denotes the median WT/MUT ratios, while the *y*-axis represents the median WT/PHOS ratios of triplicates. Notably, several SH2 domain-containing proteins exhibit negative WT/PHOS ratios, indicating preferential binding to the phosphorylated EGFR control peptide. **C** Mass spectra of an exemplary GRB2 peptide show specific GRB2 binding to the phosphorylated EGFR peptide in all three replicates. **D** Among all 111 peptide pulldowns, 70 exhibited at least one specific interactor (left), while 31 out of 37 peptide triplets had at least one specific and differential binder (right). **E** The application of two quantitative filters substantially reduced the number of interactions, with LFQ filtering revealing approximately 500 specific interactors and SILAC filtering identifying 170 specific and differential interactors.

We therefore also applied BioID to our GATAD1 variants. The results were overall consistent with the AP-MS data, with BioID yielding a higher number of interaction partners (Supplementary Fig. 41 and Supplementary Data 6). These interaction partners included several well-known members of the EMSY complex, such as EMSY, PHF12, and SIN3A[33,55] and also other described GATAD1 interactors such as ZMYND8[59]. Most importantly, we also did not observe increased interaction of 14-3-3 proteins with the phosphomimetic variant in the BioID data.

It should be noted that phosphomimetic mutations do not recapitulate all features of a phosphorylated residue[60,61]. Although, phosphomimetic mutations have been successfully employed to imitate phosphorylation in many cases[56,62–64], there are instances where they do not effectively replace phosphorylated amino acids. This is especially relevant for interactions to 14-3-3 proteins where phosphomimetic mutations mostly failed to mimic phosphorylation[65–70], except for one example[71]. The observed variations in behavior can be attributed to the contrasting biochemical properties of phosphorylated serine/threonine residues and aspartic/glutamic acid. In fact, the ability to work with phosphorylated peptides instead of phosphomimetic mutants is a key advantage of the peptide pulldown approach over genetic screens such as yeast two-hybrid and phage display[72,73].

To unambiguously test whether or not GATAD1 phosphorylated on S102 binds to 14-3-3 family proteins we turned to isothermal titration calorimetry (ITC). To this end, we expressed human recombinant 14-3-3 epsilon (YWHAE) as a GST fusion protein and assessed its interaction with various peptides (Fig. 5A). We found that GATAD1 pS102 interacted with 14-3-3ε with a binding constant of about 3 μM (Fig. 5A). This interaction strictly depends on the phosphorylation of S102: Neither the non-phosphorylated wild-type peptide nor the peptide with the disease-linked S102P mutation showed detectable binding (Fig. 5A). Importantly, the ITC data also showed that the phosphomimetic S102D and S102E mutant peptides do not interact with 14-3-3ε (Fig. 5A). Hence, phosphomimetic mutations of this phosphorylation site do not recapitulate the phosphorylated state, explaining the AP-MS and BioID data.

## Structural analysis of the GATAD1 14-3-3 interaction

The discrepancy between the phosphorylated GATAD1 and the phosphomimetic mutants renders cell biological experiments involving these mutants inconsequential. Therefore, we focused on a more detailed structural analysis of the interaction with the phosphorylated peptide instead.

First, we sought to better characterize the GATAD1 14-3-3 binding motif. The Cantley lab initially determined two 14-3-3 recognition motifs (RSXpSXP and RXF/YXpSXP) using degenerate peptide libraries[74]. Although neither of these motifs exactly matches the GATAD1 phosphorylation site, some sequence similarities are apparent (Fig. 5B). Additionally, motif prediction platforms like Scansite[75] and 14-3-3pred[76] identify this region as having the top scores for potential 14-3-3 binding, with values of 0.358 and 0.778, respectively (Fig. 5B). To experimentally determine which amino acids surrounding GATAD1-pS102 are important for 14-3-3 binding we used alanine scanning[77]. To this end, we individually replaced each amino acid surrounding the phosphorylation site by alanine in a PRISMA screen (Fig. 5C). Among the total number of 1367 proteins detected in triplicate experiments, 121 were found to differ significantly between the 15 peptides, including six 14-3-3 protein family members (14-3-3 beta/alpha, gamma, epsilon, zeta, eta and theta). Comparing the abundance of the 14-3-3 proteins across peptide pulldowns yields a number of important insights (Fig. 5C). First, all six 14-3-3 family proteins show essentially identical binding preferences. This observation is in line with recent studies indicating that 14-3-3 paralogs have very similar target-binding tendencies[78,79]. Second, all alanine-substituted phosphopeptides pulled down more 14-3-3 proteins than any of the non-phosphorylated peptides. Hence, the interaction strictly depends on phosphorylation, corroborating the ITC results. Third, we identified proline in position +2 and alanine in position +1 (substituted by glycine) as being important for binding while replacing none of the other amino acids had a significant effect. The relevance of proline at the +2 position is in good agreement with both 14-3-3 binding motifs. Also, while studies have shown that there is no strong preference for any specific amino acids in the +1 position, specific amino acids are not tolerated at this position, including glycine[74]. This explains our observation that changing the alanine at position +1 to glycine disrupts binding.

To further investigate the different binding behavior of the tested GATAD1 peptides, we used X-ray crystallography to obtain the structure of 14-3-3ε in complex with the GATAD1 phosphopeptide at 3.1 Å resolution (see Supplementary Table 1 for the complete data statistics). The two 14-3-3ε proteins in the asymmetric unit (ASU) of the crystals formed a two-fold symmetric homodimer, which is typical for the 14-3-3 protein family. Each monomer consists of nine α-helices and harbors one phosphorylated GATAD1 peptide in a binding groove formed by helices α3, α5, α7, and α9 (Fig. 5D). Superimposing the 14-3-3ε structure with the 14-3-3ζ structure bound to phosphorylated polyomavirus middle-T antigen (mT)[74] revealed a high overall structural similarity with a root mean square deviation (RMSD) of 0.87 for 421 superimposed Cα atoms.

Residues 98-107 of the GATAD1 peptide were resolved in the electron density in both 14-3-3ε molecules in the ASU, while residues 108-109 were only visible in one of them (Fig. 5D). 14-3-3ε binds the central phospho-serine 102 of the GATAD1 peptide via the triad Arg57, Arg130, and Tyr131, which is highly conserved in the 14-3-3 family. In addition, Lys50 of 14-3-3ε directly interacts with the phosphate group of the peptide (Fig. 5D). GATAD1 Pro104 packs into a hydrophobic cage formed by 14-3-3ε Leu219, Ile220, and Leu223, whereas Tyr100 of the peptide shows a T-shaped π stacking against Tyr182 and Trp231 of 14-3-3ε. Additional hydrogen bonds between the main chain of the peptide and 14-3-3 residues Asn176, Asn227, and Asp216 (only in one monomer of the ASU) contribute to the peptide orientation (Fig. 5D, bottom).

Our structure explains the deficits of the non-phosphorylated GATAD1 peptide and the pathogenic S102P and the phosphomimetic S102D peptide variants to interact with 14-3-3ε. The non-phosphorylated serine, as well as the proline chain (in the S102P variant) and the aspartic acid side chains (in the S102D peptide variant), are too short to reach into the positively charged triad patch in 14-3-3ε for proper binding (Fig. 5D, bottom). A glutamic acid side chain in the S102E mutant, on the other hand, appears to be able to reach into the

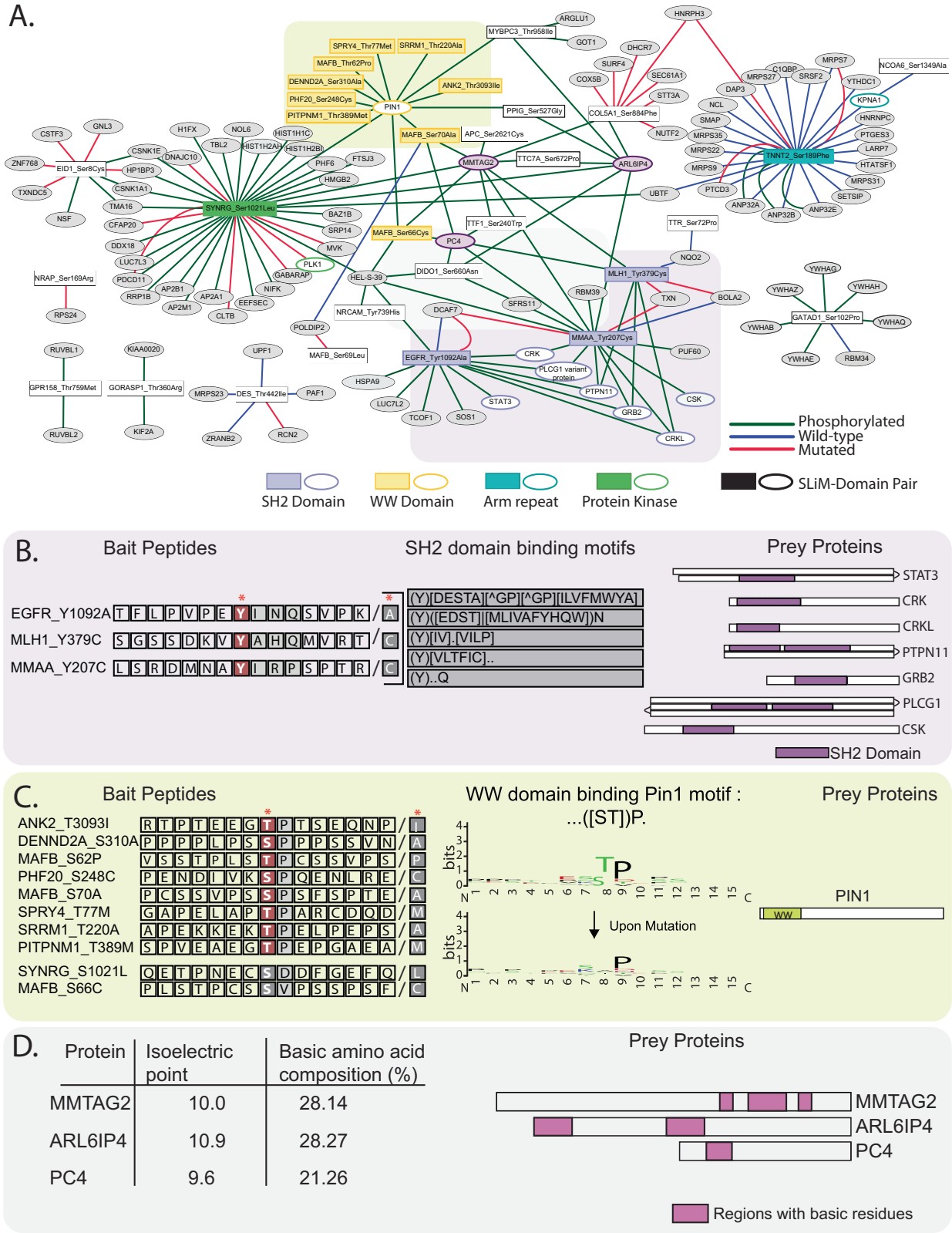

**Fig. 3 | Network and SLiM analysis. A** A Peptide-Protein network that illustrates all interactions that passed both quantitative filters (LFQ and SILAC). Rectangles represent bait peptides and ovals interacting proteins. Edges indicate preferential binding as indicated in the figure. Highlighted regions and colored nodes indicate interactions that can be explained by annotated SLiM-Domain pairs. **B** Interactions explainable by SH2 domains. Peptide sequences and corresponding SH2 domain binding SLiMs (left and middle) are displayed, along with a graphical representation of the SH2 domain-containing proteins (right). **C** Peptides binding to PIN1 protein. 8/10 peptides contain the ([ST])P motif in the phosphorylated region and bind to the WW domain of PIN1 in a phosphorylation-dependent manner. The sequence logo demonstrates the SLiM that is lost upon mutation. **D** Several peptides interact with MMTAG2, ARL6IP4 and PC4. These proteins have a compositional bias towards positively charged amino acids (right), explaining preferential binding to the more negatively charged phosphorylated peptides.

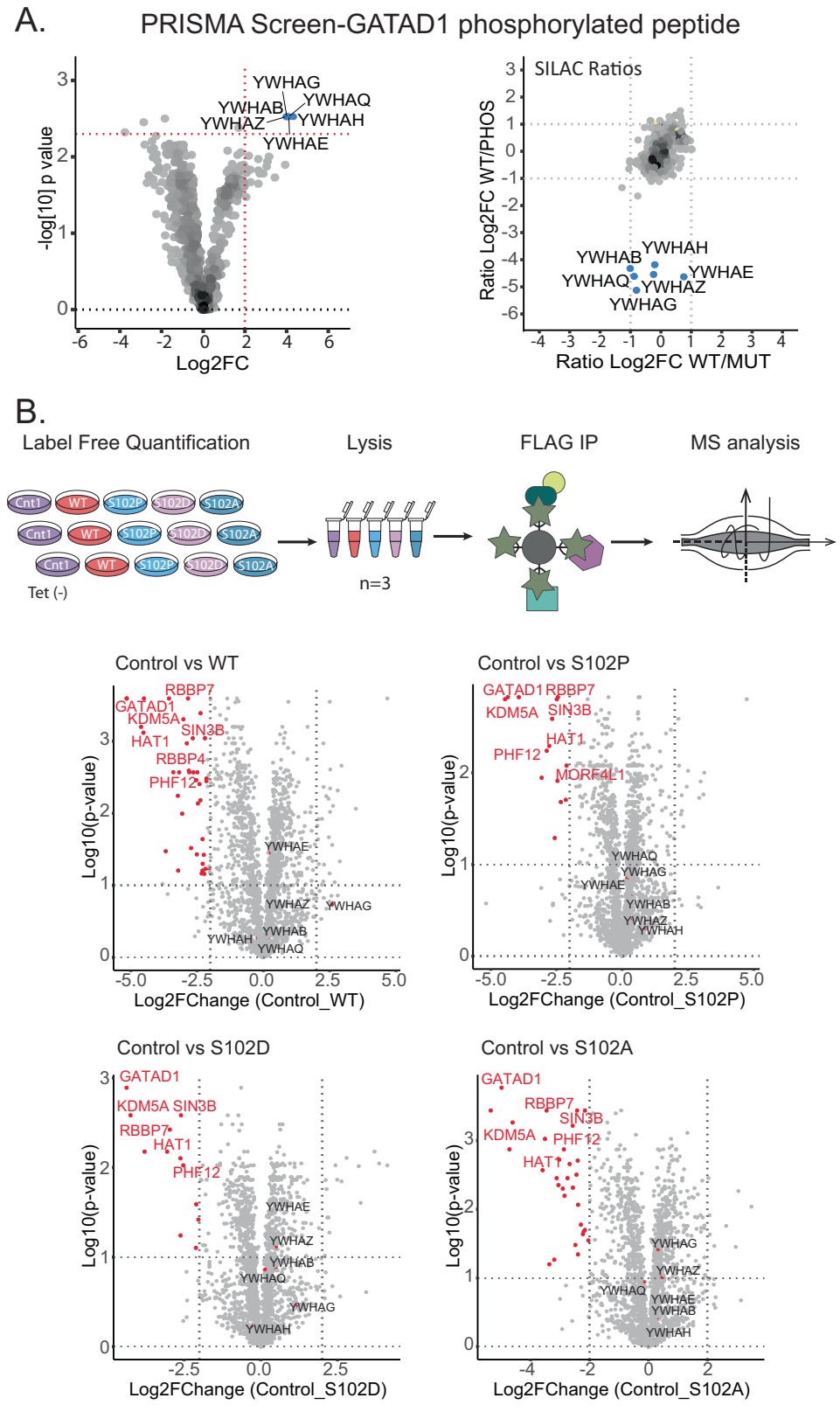

A. PRISMA Screen-GATAD1 phosphorylated peptide

B.

Control vs WT

Control vs S102P

Control vs S102D

Control vs S102A

binding triad; however, it is not able to form a hydrogen bond with Tyr131 nor Lys50 of 14-3-3ε (Fig. 5D, bottom right). In summary, our structural analyses reveal the sequence and structural requirements for GATAD1 pS102 14-3-3ε interaction, explaining why binding strictly depends on phosphorylation and why it cannot be mimicked by phosphomimetic mutations.

**The 14-3-3 binding region in GATAD1 is a NLS**

The absence of a cellular model mimicking GATAD1 phosphorylation complicates the analysis of the functional consequences of the interaction with 14-3-3 family proteins. To shed more light onto possible cellular mechanisms we took a closer look at the 121 proteins found to interact differentially to the 15 peptides in the alanine scanning

**Fig. 4 | Interaction of 14-3-3 family proteins with GATAD1 phosphorylated on serine 102. A** The phosphorylated GATAD1 peptide specifically interacts with the multiple 14-3-3 protein family members, despite lacking an annotated 14-3-3 binding motif. Both the Volcano plot (left) and the SILAC scatter plot (right) show 14-3-3 proteins as specific and differential binders of the phosphorylated GATAD1 peptide (blue: LFQ specific binders to the phosphorylated peptide). Volcano plots were generated after performing the Wilcoxon test to compare proteins identified in phosphorylated GATAD1 peptide form with all the proteins identified in all other WT and Mutant peptide forms (Significance cut-off: log2 fold change >2 and *p*-value < 0.005). **B** FLAG-IP results for full-length GATAD1. Experimental design

(Top). Volcano plots display adjusted *p*-values as a function of log2 fold changes between GATAD1 variants and the non-induced control (no tetracycline addition) following a two-sided Student t-test with Benjamin-Hochberg FDR correction (Cut-off: log2 fold change >2, *p*-values < 0.1). The data displays that all variants pull down similar proteins including the known GATAD1 interactors (protein labels in red), indicating that the mutation does not impair any of the known interactions. While 14-3-3 family proteins (YWHAB, YWHAE, YWHAH, YWHAQ, and YWHAZ) are detected, they do not show any specific interaction with any of the GATAD1 proteins.

experiment (Fig. 6A and Supplementary Data 5). Intriguingly, we observed a cluster of 72 proteins that specifically interacted with all non-phosphorylated GATAD1 peptides. Enrichment analysis of KEGG pathways and Reactome Gene Sets identified nucleocytoplasmic transport as the most significantly enriched term (Fig. 6A), with several carrier proteins facilitating nuclear localisation signal (NLS)-based nuclear import, including IPO4, IPO5, IPO7, TNPO1, TNPO3, and RANBP2. This intriguing observation suggests that this region of GATAD1 may function as a nuclear localization signal (NLS), potentially playing a role in nuclear transport. This is surprising, considering previous reports indicating that GATAD1 lacks an NLS and instead is believed to be imported via a piggyback mechanism alongside its binding partner HDAC1/2[33,80]. If the 14-3-3 protein binding region of GATAD1 was indeed an NLS, recruitment of 14-3-3 to phosphorylated GATAD1 would be expected to block nuclear import. This is reminiscent of the function of 14-3-3 proteins in multiple prior studies that have demonstrated that 14-3-3 proteins can bind to phosphorylation sites in proximity to NLSs[81–83]. This hinders recruitment of importins and thus blocks nuclear import.

To further investigate this possibility, we first used the Hidden Markov Model-based tool NLStradamus to predict possible NLSs in GATAD1[84]. Using the default settings (2 state HMM static, prediction cutoff 0.6), this algorithm indeed identified a region containing the 14-3-3 binding peptide as a potential NLS (Fig. 6B). Next, to assess if this region is a functional NLS, we created a plasmid encoding a GFP-fusion protein and transiently transfected it into HEK−293 cells. As expected, the GFP-only control showed diffuse fluorescence throughout cells. In contrast, fusing GFP to the potential GATAD1 NLS resulted in predominantly nuclear localisation (Fig. 6C). In addition, we also studied the effect of deleting the putative NLS (ΔNLS) on the subcellular localisation of GATAD1. While wild-type FLAG-tagged GATAD1 localized exclusively to the nucleus, the GATAD1 ΔNLS variant exhibited a dispersed distribution throughout the cell (Fig. 6D). We conclude that this region is both required and sufficient for the nuclear localisation of GATAD1. Hence, binding of 14-3-3 proteins to this region could indeed regulate GATAD1 trafficking.

## Discussion

Understanding the functional consequences of mutations and post-translational modifications in health and disease remains a major challenge, especially for intrinsically disordered regions. Here, we employed peptide-based interaction proteomics to investigate how known disease-associated SAVs of known serine, threonine or tyrosine phosphorylation sites affect protein-protein interactions. We identify many interactions affected by the mutation and/or the phosphorylation state (Fig. 3). We further show that a phosphorylation site in the transcription factor GATAD1 that is mutated in a family of patients with dilated cardiomyopathy binds 14-3-3 proteins (Fig. 4A). Finally, we show that the 14-3-3 binding region of GATAD1 is a functional nuclear localisation signal, suggesting that binding of 14-3-3 proteins to phosphorylated GATAD1 could affect its subcellular localisation (Fig. 6).

Although proteomics can now routinely identify thousands of phosphorylation sites, their functional characterisation is lagging

behind[9]. Recently, a number of computational[15,85] and experimental approaches[62,64,86,87] have been developed to assess the functional relevance of phosphopeptides on a more global scale. Here, we took advantage of the established PRISMA method[28–30] to study the function of phosphorylation sites coinciding with disease-associated mutations. In principle, phosphorylation can both induce and disrupt interactions. Of the 170 specific and differential peptide-protein interactions we observed, 132 were affected by the peptide phosphorylation state (see Fig. 2). Interestingly, we observed 102 phosphorylation-induced interactions while only 30 were disrupted. Hence, phosphorylation mostly tends to induce interactions. Importantly, almost all of the phosphorylation-induced interactions (101 out of 102) were disrupted by the mutations, indicating that their loss could indeed cause disease. While we observed more phosphorylation-dependent interactions in our screen, it is important to point out that mutations could also affect interactions independently of the phosphorylation state. We observed 58 interactions that are differential between the wild-type and the mutant (30 lost and 28 gained upon mutation). In this context, it is important to highlight that our screen can identify interactions that are gained upon mutation – an aspect often overlooked when investigating disease mutations (Supplementary Fig. 2B). We previously reported that pathogenic mutations can cause disease by creating dileucine motifs that lead to clathrin-binding[30]. It would be interesting to follow up on the potential interactions of the mutated forms identified here.

These numbers indicate that the modification state of a site has a broad impact on the interactome and the mutation state can highly disrupt these interactomes. This indicates that many of these mutations are pathogenic because they impair phosphorylation-dependent interactions. However, we do not know if this observation from our limited list of phosphorylation sites can be generalized. It is important to point out that not all observed differential interactions (phosphorylation- or mutation-affected) are necessarily disease-relevant. While we focused on one interaction here, our data potentially contains additional candidates that would be interesting to follow up on.

The 14-3-3 binding site of GATAD1 does not match a known 14-3-3 binding motif as defined in the ELM database[39]. We therefore extensively validated the interaction using isothermal titration calorimetry (ITC), alanine scanning and x-ray crystallography (Fig. 5). The results obtained confirm the interaction and demonstrate that despite the lack of an exact match, the interaction is overall consistent with known 14-3-3 binding patterns. This highlights the previous observation that 14-3-3 family proteins can bind to a wide range of target sequences that sometimes deviate from the canonical motifs[47,48]. More generally, our observation highlights the challenges associated with predicting SLiM-dependent interactions and the importance of experimental approaches[20,21,23].

A key advantage of the PRISMA method used here is its ability to directly compare interaction partners of phosphorylated and non-phosphorylated wild-type and mutated peptides to each other. The ability to directly utilize phosphorylated peptides is a key advantage over phosphomimetic approaches since it avoids limitations associated with charged amino acids as surrogates for phosphorylation. In

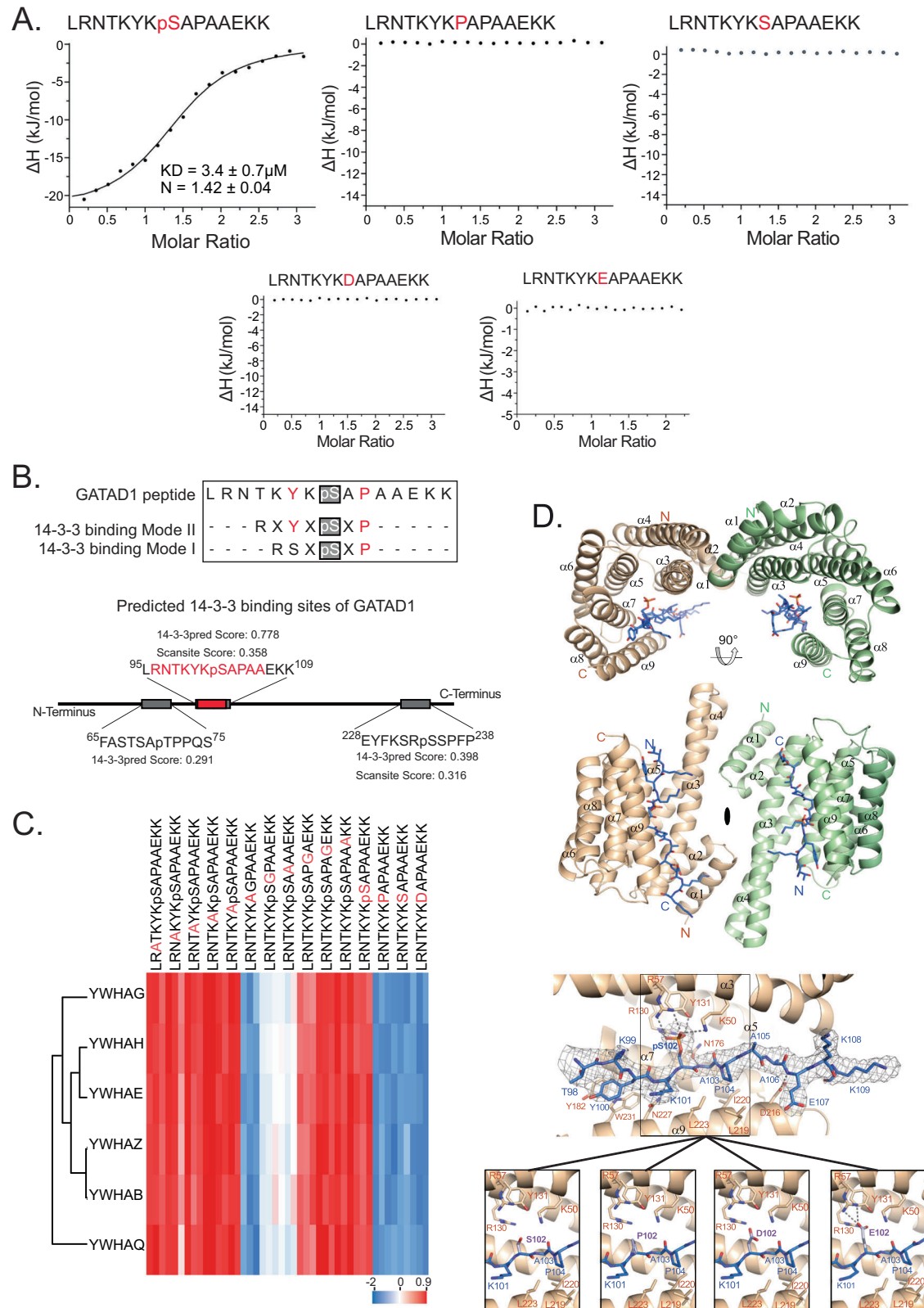

fact, we show that 14-3-3 binding to phosphorylated GATAD1 cannot be mimicked by amino acid substitution, providing another example for the problems associated with phosphomimetics[65–70].

The inability of phosphomimetic mutations to resemble the phosphorylated state and the lack of GATAD1 phosphorylation in HEK-293 cells leads to a key limitation of this study: Due to the lack of a suitable model system, we cannot fully investigate the cellular

consequences of the GATAD1 14-3-3 interaction. Moreover, we do not know which kinase(s) phosphorylate GATAD1 under which conditions. It is intriguing that phosphorylation of S102 has so far only been observed in the heart in vivo[53]. Together with the finding that the GATAD1_S102P mutation causes dilated cardiomyopathy[51], this is strongly indicative of a heart-specific function of this phosphorylation site. Additionally, our alanine scanning experiment showed that

**Fig. 5 | Structural analysis of the GATAD1 14-3-3 interaction. A** Isothermal titration calorimetry (ITC) validates the interaction of the GATAD1 phosphopeptide to 14-3-3 epsilon (YWHAE) with a Kd of 3.4 µM. In contrast, the wild-type non-phosphorylated, S102P, and phosphomimetic (S102D and S102E) mutant peptides do not interact with 14-3-3 epsilon. **B** The GATAD1 peptide displays similarities to 14-3-3 binding motifs but does not fully match. **C** Heatmap of 14-3-3 proteins quantified in a PRISMA screen with GATAD1-derived peptides modified by alanine scanning. Results highlight the importance of proline at the +2 and glycine at the +1 position. **D** Crystal structure of the 14-3-3 epsilon in complex with a phosphorylated GATAD1 peptide as determined by X-ray crystallography. Left: The side and top view of the 14-3-3 homodimer are shown as a cartoon structure with the two monomers colored green and wheat. The bound peptides are shown in stick representation in blue color. The two-fold rotation axis between the monomers is indicated by an ellipsoid in the center of the homodimer. A detailed view of the peptide binding groove with hydrogen bonds as gray dashed lines and electron densities around each peptide as a gray mesh is shown. In the bottom images, the GATAD1 phosphoserine 102 was modeled as serine, proline, aspartic acid, and glutamic acid, respectively, and highlighted in light blue.

non-phosphorylated forms of the GATAD1 peptide interact with importin carrier proteins, suggesting the mutated region plays a role in nucleocytoplasmic transport of GATAD1. Indeed, we validated that this GATAD1 region harbors a functional NLS (Fig. 6). These observations could help explain the perturbed subcellular distribution pattern of GATAD1 in cardiomyocytes of patients affected by this mutation[51]. Importantly, however, the AP-MS (Fig. 4B) and alanine scanning data (Fig. 6) indicate that the mutation itself does not affect nuclear translocation. Building upon existing data for the role of 14-3-3 proteins in nucleocytoplasmic transport[81–83,88], we propose that the binding of 14-3-3 to GATAD1 masks the NLS and impairs the protein's nuclear localization. Nevertheless, due to the lack of a cellular model system, we are not able to experimentally validate this hypothesis.

Overall, our study highlights the potential of the PRISMA screen for elucidating the functional consequences of mutations and PTMs, advancing our knowledge of protein regulation and interaction networks. In the future, it will be interesting to also investigate mutations adjacent to phosphorylation sites, not just changes of the phosphorylated residue itself. Such mutations can also modify SLiMs and thereby change PPIs. For example, the lung cancer associated P1019L mutation in EGFR has already been shown to switch the binding specificity of the adjacent phosphorylation site pY1016[32]. Also, a recent study showed that phosphorylation often modulates affinities of interactions when occurring in motif flanking regions[72]. Combining PRISMA with ultrahigh throughput proteomics enables the analysis of a much larger number of sites[89–91]. Furthermore, exploring a broader spectrum of sites and incorporating various types of modifications could reveal previously unknown domain-motif relationships.

## Methods

### Cell culture
HEK-293 (DSMZ Cat#ACC635), HEK-293T (DSMZ Cat#ACC305), and Flp-In™-293 T-REx (Thermo Fisher Scientific, R78007) cells were cultured in DMEM medium supplemented with 10% fetal calf serum (FCS) from Pan-Biotech. The cells were incubated at 37 °C with 5% CO2.

For SILAC labeling, SILAC DMEM from Life Technologies was used. The SILAC DMEM was supplemented with 10% dialyzed FCS from Pan-Biotech, glutamine (Glutamax, Life Technologies), and non-essential amino acids. Different SILAC formulations were employed: Arg0 and Lys0 for light labeling, Arg6 and Lys4 or only Lys4 for medium-heavy labeling, or Arg10 and Lys8 (Sigma-Aldrich) for heavy labeling. To ensure complete incorporation of SILAC amino acids, cells were passaged and grown for at least 2 weeks or approximately 8 doublings.

### Selection of peptide candidates for PRISMA
Pathogenic mutations affecting phosphorylation sites were selected from the PTMVar dataset in PhosphositePlus (Hornbeck et al.[36]). The PTMVar dataset provides comprehensive information on Post-Translational Modifications (phosphorylation, ubiquitylation, acetylation, methylation, and succinylation) that overlap with genetic variants associated with diseases and genetic polymorphisms.

In the database, mutations are categorized as polymorphism, unclassified, or disease. Mutations labeled as 'disease' are those associated with Mendelian diseases, while somatic cancer mutations are designated as 'disease' only if observed in three to five distinct cancer patients[36].

For our analysis, we specifically focused on entries in which mutations were causative for disease (VAR_TYPE = Disease) resulted in changes to phosphorylation sites (MOD_TYPE = Phosphorylation), and directly altered the phosphorylated amino acid (VAR_POSITION = 0). We further filtered the entries to include only peptides originating from disordered regions and with no additional reported PTMs on the peptide (peptides with more than one lowercase letter in the MOD-SITE_SEQ column were excluded). Disorder was predicted using IUPred[92], using 'SHORT' profile and considering a neighborhood of 25 amino acids. Regions with an IUPred score higher than 0.5 are considered disordered. In cases where the residue of interest mutated into multiple amino acids, we retained only one mutant form. Specifically, we excluded amino acid substitutions that could still be phosphorylated (Ser, Thr, Tyr). Following these criteria, we selected a total of 38 variant peptides for further analysis (see Supplementary Data 1). As a positive control, we included an EGFR peptide known to contain an SH2 domain binding motif[31].

### PRISMA experimental setup
A total of 117 peptides, consisting of 15 amino acids each, were synthesized in situ on a cellulose membrane using SPOT synthesis techniques[25] provided by JPT Peptide Technologies in Berlin, Germany. Among these peptides, 39 were wild-type non-phosphorylated, 39 were mutated, and 39 were phosphorylated. Whenever possible, the mutation site was positioned at position 7, which is in the center of the immobilized peptides, with their C-termini serving as the point of immobilization.

For the experimental procedure, three membranes were initially incubated with the lysis buffer (HEPES (50 mM, pH 7.9), NaCl (150 mM), EGTA (1 mM), MgCl2 (1 mM), glycerol (20%), NP-40 (1%), SDS (0.1%), and sodium deoxycholate (0.5%)). The membranes were incubated with the lysis buffer until completely moistened. To minimize non-specific binding, the membranes were then treated with yeast t-RNA at a concentration of 1 mg/ml for 10 minutes at 4 °C, following two 5 min washes with the lysis buffer.

HEK-293 cells were lysed with the lysis buffer and the protein concentration was measured using DC protein assay (Bio-Rad). Subsequently, the membranes were incubated with HEK-293 cell lysate (5 ml at a concentration of 8 mg/ml) labeled as heavy, medium, or light SILAC, and this incubation step was carried out for 2 hours at 4 °C. Afterward, the membranes were washed twice for 5 min with a washing buffer containing HEPES (50 mM, pH 7.9), NaCl (150 mM), EGTA (1 mM), and MgCl2 (1 mM). Finally, the membranes were air dried to complete the procedure.

### PRISMA sample preparation and LC/MS analysis
Following the drying of the membranes, the peptide spots were carefully excised using a 2 mm diameter ear punch (Carl Roth). To facilitate further analysis, SILAC triplets were combined in a single 96-well plate, with each well containing 30 µl of denaturation buffer composed of 6 M urea (Sigma-Aldrich), 2 M thiourea (Sigma-Aldrich), and 10 mM HEPES at pH 8. The samples underwent reduction by adding 10 mM DTT (Sigma-Aldrich) and incubating for 30 minutes at room

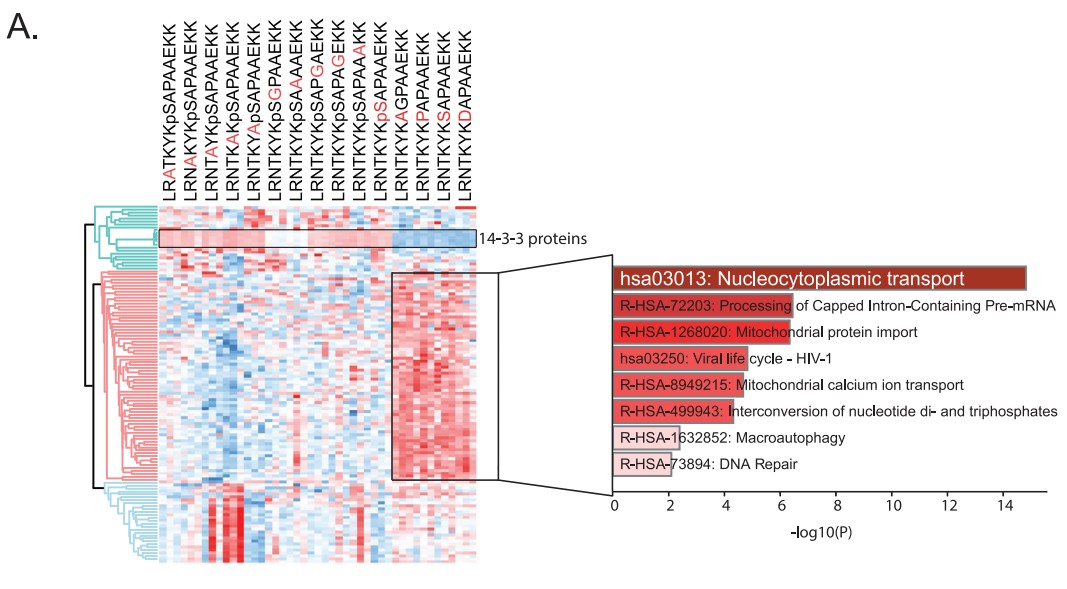

temperature (RT). Subsequently, an alkylation step was performed by adding 10 μl of 50 mM iodoacetamide (IAA) (Sigma-Aldrich) to each well and incubating for an additional 30 min at RT in the dark.

To initiate digestion, 0.5 μg of LysC enzyme was added to each well, and the samples were incubated for 1.5 h at RT. Subsequently, the digestion was continued with 0.5 μg of trypsin (Promega) overnight at RT. The digestion process was halted the next day by acidifying the samples with 10 μl of 10% trifluoroacetic acid (TFA). Finally, the samples were desalted using the standard StageTip method[93].

The elution of samples from StageTips was carried out using buffer B, consisting of 0.5% acetic acid and 80% acetonitrile. Subsequently, the eluted samples were dried using a speedvac (Eppendorf) and resuspended in buffer A, which contained 0.5% acetic acid, prior to analysis. For nano-scale reversed-phase liquid chromatography

**Fig. 6 | The 14-3-3 binding region of GATAD1 harbors a nuclear localisation signal (NLS). A** Heatmap of significant proteins identified in the alanine scanning experiment after a multiple-sample one-way ANOVA test with a permutation-based FDR of 0.05. The largest protein cluster comprises 72 proteins that exhibit specific binding to the non-phosphorylated peptide forms. Enrichment analysis of KEGG pathway and Reactome Gene Sets of these proteins identified nucleocytoplasmic transport as the most significantly enriched term. Each peptide pull-down was performed in triplicates and all three replicates are represented in the heatmap. The cluster of 14-3-3 family proteins is also highlighted (see Fig. 5C for more details). **B** The NLS of GATAD1 as predicted by NLStradamus. The GATAD1 peptide studied in this paper is highlighted in red. **C** Immunofluorescence studies of GFP and GFP-fused to the putative GATAD1 NLS show predominant nuclear localisation of the fusion protein while the GFP only signal is widespread. $n = 5$ and $n = 4$ separate images with several cells each were used to quantify GFP-control and GFP-GATAD1 peptide signals respectively using ImageJ. **D** Immunofluorescence studies comparing FLAG-GATAD1 wild-type with FLAG-GATAD1 ΔNLS (lacking the putative NLS) reveal distinct localization patterns: while the wild-type GATAD1 is predominantly nuclear, the ΔNLS variant exhibits a widespread distribution, with approximately half of the protein found in the cytoplasm. ImageJ was used for signal quantification with $n = 12$ individual images (several cells each) per condition. In both (**C** and **D**), the $p$-values were calculated by performing a two-sided Welch's $t$-test. Scale bar = 50 μm. Signal intensities used to perform the calculations are provided as source data.

coupled with tandem mass spectrometry (nanoLC/MS/MS), an Orbitrap Fusion mass spectrometer (Thermo Fisher Scientific) was utilized. The mass spectrometer was connected to a Thermo Ultimate 3000 RSLCnano pump equipped with a self-pulled analytical column measuring 150 mm in length and 100 μm in internal diameter. The analytical column was packed with ReproSil-Pur C18-AQ material (3 μm; Dr. Maisch GmbH). The mobile phases consisted of buffer A (0.5% acetic acid) and buffer B (0.5% acetic acid and 80% acetonitrile)[94].

During the analysis, peptides were eluted from the analytical column at a flow rate of 500 nl/min, employing a gradient as follows: 5 to 10% B over 5 min, 10 to 40% B over 60 min, 40 to 99% B over 5 min, and maintaining 99% B for 5 min. The samples were measured in the orbitrap fusion mass spectrometer using the following methods: a full scan was performed with a resolution of 120,000 over an $m/z$ range of 300–1500. The maximum injection time was set to 50 ms, and the AGC target was set at 400,000. Following the full scan, 20 MS/MS scans were conducted using isolation mode with a quadrupole, an isolation window of 1.6, HCD activation type, 30% collision energy, IonTrap detector type, and a maximum injection time of 35 ms.

### PRISMA data analysis

The analysis of the raw files was performed using MaxQuant version 1.6.2.6a[95], with default settings except for enabling the match between run and requantify options. Variable modifications were set to oxidation of methionines and acetylation of N-terminal residues, while fixed modifications were set to Carbamidomethylation. In silico digestion of proteins in the reference database, uniprot_human_20181012_canonical_isoform.fasta, and peptide identification were carried out using trypsin/P, allowing a maximum of 2 missed cleavages. The data were analyzed in both SILAC mode, with Lys4 as the medium-heavy label and Lys8 and Arg10 as the heavy labels, as well as in label-free quantification (LFQ) mode. A false discovery rate of 1% was set at both the peptide-spectrum matches (PSM) and protein levels, and the assessment was performed by searching a decoy database generated by reversing the reference database.

The resulting MaxQuant files were further analyzed using R (R version 4.2.1 and Rstudio version 2022.07.1), including all statistical analysis and generation of figures. Figures were modified using Adobe Illustrator CS6. Initially, the ProteinGroups table was filtered to exclude proteins identified only by site, proteins from the reverse database, and potential contaminants. Peptide candidates identified based on their tryptic sites were also filtered out. LFQ values of the remaining proteins were log2 transformed and filtered to retain only those with at least one valid value within the three replicates. Missing values were imputed by randomly drawing values from a log distribution calculated as 0.25 times the standard deviation of the measured log-transformed values, down-shifted by 1.8 standard deviations. These LFQ values were used to determine peptide-specific interactors. The nonparametric Wilcoxon test was employed to compare the median LFQ values of the proteins identified in triplicates ($n = 3$ biological replicates) of a peptide with the median LFQ values of the background proteins. The background comprised proteins identified in all other pull-downs, excluding the corresponding variant peptide. Due to the charge effect, plenty of phosphorylated peptide variants interacted with the same proteins. We saw it fit to keep these proteins in the analysis. Therefore, all phospho-pull-downs were excluded from the background. The resulting $p$-values and fold changes were plotted as volcano plots. The significance cut-off was established using the control EGFR_pTyr1092 peptide, previously characterized as SH2 domain binding. A cut-off was set to capture all SH2 domain proteins identified with the phosphorylated peptide variant, requiring a log2 fold change greater than 2 and a $p$-value smaller than 0.005.

The SILAC ratios were then utilized to determine the differential interactors. The SILAC ratios were logarithmized, normalized by subtracting the median SILAC ratio of each experiment from all individual SILAC ratios in that experiment, and subjected to label swap. This resulted in triplicates of triple SILAC experiments (wt/mut, wt/phos, and phos/mut ratios). The SILAC data were filtered to retain only those with at least two valid values out of the three replicates, and the medians of these ratios were plotted on a scatter plot with wt/mut on the $x$-axis and wt/phos on the $y$-axis. Differential proteins were identified as those with wt/mut, phos/mut, and/or wt/phos ratio log2 fold change greater than one or smaller than minus one. The specific and differential proteins obtained are summarized in Supplementary Data 2 and were used to create the peptide-protein network using Cytoscape v.3.9.1[96].

### SLiM-domain global analysis of PRISMA results

For global SLiM-Domain analysis, ELM classes and ELM-PFAM interactions were downloaded from the ELM database. Protein PFAM domains were downloaded from UniProtKB. Afterward, the PFAM domains were matched to the proteins identified as interactors and motifs were matched to the peptides used as baits.

For each protein, the mean LFQ intensity was calculated (within three replicates), and the missing values were converted to the minimum LFQ value observed in the screen. The LFQ intensities were $z$-scored per protein and the data was separated into two groups for each peptide form: SLiM-Domain pairs and others. The distribution of the normalized LFQ intensities between these groups was plotted in Supplementary Fig. 1C.

### Generation of HEK-293 cell lines expressing GATAD1 variants

The GATAD1 gene variants were ordered in a pTwist ENTR Kozak vector from Twist Bioscience. The genes were then transferred to a pDEST_pcDNA5_BirA_FLAG_Nterm vector[97] using the Gateway Clonase II system (Thermo Fisher Scientific). The resulting vectors, pEXPR_-GATAD1-WT_N-term_BirA_FLAG, pEXPR_GATAD1-S102P_N-term_BirA_-FLAG, pEXPR_GATAD1-S102D_N-term_BirA_FLAG, and pEXPR_GATAD1-S102A_N-term_BirA_FLAG, were used to generate tetracycline-inducible stable Flp-In^TM-293 T-REx cells. A 6-well plate was co-transfected with 0.5 μg of the pEXPR plasmid and 1 μg of the pOG44 Flp-recombinase expression vector (Thermo Fisher Scientific) using 4.5 μg of PEI (Polysciences) as a transfection reagent. The following day, the cells were trypsinized and transferred to a 10 cm dish

containing DMEM supplemented with 10% FCS and 200 µg/ml hygromycin B (Invivogen) for selection. The cells were cultured for approximately 18 days with media exchange every 3 days until visible colonies were observed. After around 18 days, the colonies were trypsinized and transferred to another 10 cm dish for further characterization and experiments.

## Interactome analysis of GATAD1 variants using BioID

Stable Flp-In™-293 cell lines expressing GATAD1 variants were cultured using SILAC light (Lys0, Arg0) and SILAC heavy (Lys8, Arg10) media, as explained in the cell culture section. After full labeling, cells were split into 15 cm dishes and were treated with 1 µg/mL tetracycline for 24 h to induce the expression of GATAD1. After the induction period, cells were incubated with 50 µM biotin overnight for proximity biotinylation. As a control, wild-type GATAD1 cells, without biotin or tetracycline addition, were used. Samples were multiplexed, as shown in Supplementary Fig. 41A with both forward and reverse label swap ($n = 2$). Cells were then lysed in lysis buffer (50 mM Tris-HCl pH 7.5, 150 mM NaCl, 1% Triton X-100, 1 mM EDTA, 1 mM EGTA, 0.1% SDS) with freshly added protease inhibitors and 1% sodium deoxycholate. To digest the excess DNA, 1 µl of Benzonase was added, and the samples were incubated for 20 min at 37 °C. Biotinylated proteins were enriched for 3 h at 4 °C using streptavidin-sepharose beads (GE Cat# 17-5113-01). Beads were pre-washed twice with 0.01% BSA and twice with lysis buffer. After enrichment, the beads were washed one time with lysis buffer, two times with washing buffer (50 mM HEPES-KOH pH 8.0, 100 mM KCl, 10% glycerol, 2 mM EDTA, 0.1% NP-40), and six times with 50 mM ammonium bicarbonate. Beads were resuspended in ammonium bicarbonate, and an on-bead protein digest with 1 µg of trypsin followed overnight. The next day, the digested proteins were transferred to a fresh tube and were incubated with 10 mM DTT for 30 min at 37 °C and with 55 mM iodoacetamide for 20 min at 37 °C in the dark. Samples were desalted with StageTips.

Peptides were separated using reversed-phase liquid chromatography (EASY nLC II 1200, Thermo Fisher Scientific) with self-made C18 microcolumns (20 cm long) packed with ReproSil-Pur C18-AQ 1.9 µm resin (Dr. Maisch, cat# r119.aq.0001). The chromatography system was coupled online to the electrospray ion source (Proxeon) of an Orbitrap Exploris 480 mass spectrometer (Thermo Fisher Scientific). The mobile phase consisted of buffer A (0.1% formic acid and 5% acetonitrile) and buffer B (0.1% formic acid and 80% acetonitrile). Peptides were eluted using a gradient with increasing concentrations of buffer B over 110 minutes, at a flow rate of 250 nl/min. Mass spectrometry data was acquired in data-dependent mode with settings for one full scan (resolution: 60,000; *m/z* range: 350–1600; normalized AGC target: 300%; maximum injection time: 10 ms), followed by top 20 MS/MS scans using higher-energy collisional dissociation (resolution: 15,000; *m/z* range: 200–2000; normalized AGC target: 100%; maximum injection time: 120 ms; isolation width: 1.3 *m/z*; normalized collision energy: 28%).

Raw files were analyzed with MaxQuant, version 2.0.3.0. The Protein Groups data table was further processed and analyzed using R (R version 4.2.1 and Rstudio version 2022.07.1). The table was first filtered for contaminants, identified only by site and identified by the reverse database. The SILAC ratios were log2 transformed and the labels were swapped. The data was visualized in scatter plots with the forward SILAC ratios on the x-axis and reverse SILAC ratios on the y-axis. Differential proteins were identified as those with a ratio log2 fold change greater than one or smaller than minus one (Supplementary Data 6).

## Interactome analysis of GATAD1 variants using AP-MS

Stable Flp-In™-293 cell lines expressing GATAD1 variants were cultured in DMEM supplemented with 10% FCS. 24 h before the experiment the cells were induced with 1 µg/mL tetracycline. One 15 cm dish

of cells was used per replicate and three replicates were used per condition ($n = 3$ biological replicates). As a control, a mixture of non-induced cells originating from all the variants was used. The cells were lysed using 600 µl of lysis buffer (50 mM Tris HCl, pH 7.4, with 150 mM NaCl, 1 mM EDTA, and 1% TRITON X-100)

For FLAG immunoprecipitation (IP) ANTI-FLAG M2 Magnetic Beads (M8823, SIGMA-ALDRICH) were used following the manufacturer's protocol. In short, 40 µl of the 50% bead suspension was used per reaction (replicate). The beads were first washed twice with TBS (50 mM Tris HCl, 150 mM NaCl, pH 7.4) buffer, then the cell lysate was added on top of the beads. The immunoprecipitation reaction took place for 2 hours rotating gently. After 2 h the reaction tubes were placed on a magnetic rack for separation. The supernatant was discarded while the beads were washed three times with TBS. Beads were resuspended in ammonium bicarbonate, and an on-bead protein digest with 1 µg of trypsin followed overnight. The next day, the digested peptides were transferred to a fresh tube and were incubated with 10 mM DTT for 30 minutes at 37 °C and with 55 mM iodoacetamide for 20 minutes at 37 °C in the dark. Samples were desalted with StageTips and were eluted from there with a buffer containing 50% acetonitrile and 0.1% formic acid, they were dried, and resuspended in a solution containing 3% acetonitrile and 0.1% formic acid (Buffer A).

Peptides were separated using reversed-phase liquid chromatography (EASY nLC II 1200, Thermo Fisher Scientific) with self-made C18 microcolumns (20 cm long) packed with ReproSil-Pur C18-AQ 1.9 µm resin (Dr. Maisch, cat# r119.aq.0001). The chromatography system was coupled online to the electrospray ion source (Proxeon) of an Orbitrap Exploris 480 mass spectrometer (Thermo Fisher Scientific). The mobile phase consisted of buffer A (0.1% formic acid and 5% acetonitrile) and buffer B (0.1% formic acid and 80% acetonitrile). Peptides were eluted using a gradient with increasing concentrations of buffer B over 45 min, at a flow rate of 250 nl/min. Mass spectrometry data was acquired in data-dependent mode with settings for one full scan (resolution: 60,000; *m/z* range: 350–1600; normalized AGC target: 300%; maximum injection time: 10 ms), followed by top 20 MS/MS scans using higher-energy collisional dissociation (resolution: 15000; *m/z* range: 200–2000; normalized AGC target: 100%; maximum injection time: 22 ms; isolation width: 1.3 *m/z*; normalized collision energy: 28%).

The raw files were further analyzed using MaxQuant version 1.6.7.0 in a label free mode. The resulting ProteinGroups table was further processed with R (R version 4.2.1 and Rstudio version 2022.07.1). The table was filtered for potential contaminants, identified only by site and identified by the reverse database. The data was then log2 transformed and filtered for at least two valid values in at least one of the triplicates. Leftover missing values were imputed in the same way as in the PRISMA screen analysis. A Student *t*-test with Benjamin Hochberg FDR correction was then employed to compare different conditions with the control. A significance cut-off of a log2 fold change >2 and a *p*-value < 0.1 was selected (Supplementary Data 4). The results were visualized as volcano plots.

## Recombinant protein expression and purification of YWHAE

The 14-3-3ε (YWHAE) cDNA was purchased from Twist Bioscience in the pTwist Chlor High Copy vector and cloned into the pGEX6P1 plasmid (GE Healthcare) for recombinant expression as a GST-fusion protein followed by a Prescission protease cleavage site. The plasmid was freshly transformed into *Escherichia coli* C41 (DE3) cells. Cultures were grown in terrific broth supplemented with ampicillin (100 µg/ml) at 37 °C and 80 rpm until an optical density at 600 nm of 0.7 was reached. Protein expression was subsequently induced by the addition of 300 µM isopropyl β-d-1-thiogalactopyranoside (IPTG), and the cultures were grown for another 18 hours at 20 °C. Cells were harvested by centrifugation at 4000 g and frozen at −20 °C.

Following resuspension in lysis buffer (50 mM HEPES/NaOH pH 7.5, 500 mM NaCl and 3 mM Dithiothreitol (DTT)) supplemented with 1 mg deoxyribonuclease I (DNaseI, Roche) and protease inhibitor 4-(2-Aminoethyl) benzenesulfonyl fluoride hydrochloride S7(AEBSF), cells were disrupted using a microfluidizer (Microfluidics). The cell extract was centrifuged at 55,000 g for 45 min at 4 °C to remove insoluble parts. The cleared supernatant was applied onto a prepacked Glutathione Sepharose 4B column (Cytiva) equilibrated in the lysis buffer. The column was extensively washed with lysis buffer. To cleave off the GST tag, 1 mg PreScission protease was diluted in 5 ml OCC buffer (50 mM HEPES/NaOH pH 7.5, 150 mM NaCl, and 3 mM DTT), applied to the column material, and incubated overnight at 4 °C. Cleaved 14-3-3ε protein was eluted from the column using OCC buffer, concentrated and loaded onto an 16/60 S200 column (Cytiva) equilibrated in SEC buffer (20 mM HEPES/NaOH pH 7.5, 150 mM NaCl and 2 mM DTT). Pure and homogenous 14-3-3ε was concentrated to 13 mg/ml, flash-frozen in liquid nitrogen, and stored at −80 °C until further use.

### Isothermal titration calorimetry (ITC)

GATAD1 peptides used in these experiments were obtained from JPT peptide technologies with an N-terminus in amine and a C-terminus in amide form with 95% purity. ITC experiments were performed using a PEAQ-ITC microcalorimeter (Malvern). All titrations were performed at 18 °C with 400 μM peptide in the syringe and 25 μM 14-3-3ε in the reaction chamber. The protein and the titration components were dissolved in a buffer containing 20 mM HEPES pH 7.0, 150 mM NaCl, and 2 mM DTT. Malvern software was used for data visualization and fitting. The raw ITC results can be found in Supplementary Fig. 42.

### Crystallization and structure determination

The protein 14-3-3ε at a concentration of 13 mg/ml in 20 mM HEPES pH 7.5. 150 mM NaCl, 2 mM DTT were combined with the GATAD1 peptide 95-LRNTKYKpSAPAAEKK-109 in 1:2 molar ratio for complex formation. Crystallization setups were performed with the sitting-drop vapor diffusion method by using a Gryphon pipetting robot (Matrix Technologies Co.) for pipetting 200 nl of protein to an equal volume of precipitant solution. The Rock Imager 1000 storage system (Formulatrix) was used for storing and imaging of the experiments. Crystals appeared within 3–10 days in 19% PEG 3350, 0.35 M NaBr, 0.1 M BisTris-Propane pH 6.5 at 20 °C and were flash-frozen in liquid nitrogen in the presence of 20% ethylene glycol. Diffraction data were collected on BL14.1 at the BESSY II electron storage ring operated by the Helmholtz-Zentrum Berlin[98], processed and scaled using XDSapp[99]. The 3.2 Å structure was solved by molecular replacement with Phaser[100] using the 14-3-3 protein epsilon structure (PDB: 2BR9 [https://doi.org/10.2210/pdb6ynv/pdb]) as a search model. The structure was built using COOT[101] and iteratively refined with Refmac[102]. Data statistics are summarized in Supplementary Table 1. Crystals of 14-3-3 epsilon with the GATAD1 phosphopeptide belong to space group R32 containing 2 complex molecules per asymmetric unit connected by a two-fold rotational symmetry. Residues 2-235 of 14-3-3 are explained in the electron density, whereas residues 1 and 236-255 are disordered and therefore not visible in the electron density. The GATAD1 peptide is covered by residues 98–109 in chain C and residues 98–107 in chain P. 97.0% of the residues in the complex structure were in the favored regions and no outlier was observed in the Ramachandran map. The Ramachandran statistics were analysed using Molprobity[103]. Figures were generated with PyMol (http://www.pymol.org). The atomic coordinates of 14-3-3ε with the GATAD1 95-LRNTKYKpSAPAAEKK-109 peptide structure have been submitted to the Protein Data Bank with the entry code 8Q1S.

### Alanine scanning

To investigate the contribution of individual amino acids in the GATAD1 peptide for binding to 14-3-3 proteins, an alanine scanning experiment was designed. In this experiment, each amino acid from the −5 to +5 positions surrounding the phosphorylated residue was mutated to alanine, except for original alanine residues which were mutated to glycine. The experiment also included the wild-type phosphorylated, wild-type non-phosphorylated, disease-associated mutant, and aspartic acid mutant peptides. These 15 peptides were synthesized on a cellulose membrane in triplicates ($n = 3$ biological replicates) to facilitate the pull-down of proteins from HEK-293T cell lysate. The synthesis of peptides, pull-downs, and sample preparation for mass spectrometry followed the procedures described earlier for the PRISMA screen. Subsequently, samples were eluted from StageTips using a solution containing 50% acetonitrile and 0.1% formic acid, dried, and resuspended in a solution containing 3% acetonitrile and 0.1% formic acid. LC-MS measurements were performed similarly as in the AP-MS experiment except for the gradient, which this time was 45 min, and the maximum injection time for MS2 scans was 22 ms.

The acquired mass spectra were subjected to further analysis using MaxQuant (version 1.6.3.4). Subsequent analysis was performed using Perseus (version 1.6.7.0). The ProteinGroups data table was filtered to exclude potential contaminants, proteins identified only by site, and proteins identified by the reverse database. LFQ values of the remaining proteins were log2 transformed, and only proteins with at least two valid values in three replicates were considered for analysis. An one-way ANOVA multiple-sample test with a permutation-based FDR of 0.05 was conducted. The LFQ values of significant proteins were z-scored by row and hierarchical clustering was performed (Fig. 6A and Supplementary Data 5). To better visualize the 14-3-3 proteins, their LFQ values were extracted from the whole table and were z scored and clustered individually (Fig. 5C).

### Subcellular localization of GATAD1 variants

Flp-In™-293 cells stably expressing WT-GATAD1 were cultured and maintained. Cells were seeded on glass coverslips at a density of 50,000 cells per 24-well plate. The following day, tetracycline was added to a final concentration of 1 μg/mL, and the cells were incubated for an additional 24 hours. After the incubation period, the cells were fixed with 4% PFA for 15 min. Subsequently, the cells were washed three times for 5 min each with PBS. Permeabilization was achieved by incubating the cells with 0.5% Triton X-100 in PBS for 10 min, followed by two washes with PBST (PBS with 0.1% Triton X-100). To reduce the background signal, the cells were incubated with a blocking solution (1.5% BSA in PBST) for 1 h. Next, the cells were incubated with the primary antibody, mouse monoclonal IgG anti-GATAD1 (sc-81092, 1:100, Santa Cruz), in a blocking solution for 1 h. After washing the cells three times for 5 min with PBST, the cells were incubated with the secondary antibody, goat anti-mouse IgG (H + L) Alexa Fluor 488 (A11008, 1:500, Invitrogen), and phalloidin - Alexa 594 (A12381, 1:500, Invitrogen) for 1 h. Following another round of washing with PBST, the cells were incubated for 3 min with 0.1 μg/mL DAPI (Sigma Aldrich) in PBS. A final wash with PBS was performed, and the coverslips were then washed in MilliQ water and mounted on slides using ProLong Gold Antifade Mountant (Life Technologies). All images were acquired using a Leica DM5000B microscope with an HC PL APD 63x/1.4-0.6 objective and Leica Application Suite X software. The acquired images were processed using Fiji ImageJ software[104].

### Transfection of HEK-293T with GFP fused GATAD1 peptide

The pTwist_CMV_GFP_SPACER_KQSKQEIHRRSARLRNTKYKSAPAAEKK VSTKGKGRR constructs were ordered in the pTwist_CMV expression vector from TwistBioscience (https://www.twistbioscience.com).

HEK-293T cells were seeded on glass coverslips at a density of $0.1 \times 10^6$ cells per 12-well plate. The following day, cells were transfected with 1 μg of the pTwist_CMV plasmid using 3 μg of PEI (Polysciences) as a transfection reagent or with 1 μg of the pDEST_GFP plasmid (Control).

## Transfection of HEK-293T with ΔNLS FLAG-GATAD1

pTwist_CMV_FLAG_Spacer_GATAD1_wild-type and pTwist_CMV_FLAG_Spacer_GATAD1_ΔNLS (position 82–118 deleted) constructs were ordered in the pTwist_CMV expression vector from TwistBioscience. HEK-293T cells were seeded on glass coverslips at a density of $0.1 \times 10\string^6$ cells per 12-well plate, 24 h later were transfected with 1 μg of the pTwist_CMV (full length or ΔNLS) plasmids using effectene transfection reagent.

## Immunofluorescence studies on HEK-293T cells

24 hours after transfection, the cells were fixed with 4% PFA for 15 min. Following fixation, cells were washed three times for 5 min each with PBS. Permeabilization was performed by incubating the cells with 0.5% Triton X-100 in PBS for 10 min, followed by two washes with PBST (PBS with 0.1% Triton X-100). For the GFP only and GFP-GATAD1 peptide experiment, the cells were incubated with 0.1 μg/mL DAPI (Sigma Aldrich) in PBS for 3 min. After another wash with PBS, the coverslips were rinsed in MilliQ water and mounted on slides using ProLong Gold Antifade Mountant (Life Technologies).

For FLAG-GATAD1 WT and FLAG-GATAD1 ΔNLS experiment after permeabilization the cells were blocked with a blocking solution (1.5% BSA in PBST) for 1 h. Next, the cells were incubated with the primary antibody, mouse monoclonal IgG anti-GATAD1 (sc-81092, 1:100, Santa Cruz), in a blocking solution for 1 h. After washing the cells three times for 5 min with PBST, the cells were incubated with the secondary antibody, goat anti-mouse IgG (H + L) Alexa Fluor 488 (A11001, 1:500, Invitrogen), and phalloidin - Alexa 594 (A12381, 1:500, Invitrogen) for 1 h. Following another round of washing with PBST, the cells were incubated for 3 min with 0.1 μg/mL DAPI (Sigma Aldrich) in PBS. A final wash with PBS was performed, and the coverslips were then washed in MilliQ water and mounted on slides using ProLong Gold Antifade Mountant (Life Technologies).

All images were acquired using a Leica DM5000B microscope with an HC PL APD 63x/1.4-0.6 objective and Leica Application Suite X software. The acquired images were processed using Fiji ImageJ software[104].

In the manual quantification of GFP, GFP-GATAD1 peptide, FLAG-WT GATAD1, and FLAG-ΔNLS GATAD1 nuclear and cytoplasmic signals (Fig. 6C, D), we analyzed 4, 5, 12, and 12 individual immunofluorescence images, respectively. The DAPI channel was utilized to create a nuclei mask, which was used to separate the images into nuclear and cytoplasmic segments. The intensities of these segments were measured, and the mean intensities were utilized for further calculations. Statistical analysis was performed using a two-sided Welch's t-test, and the corresponding p-values are indicated in the plot.

## Reporting summary

Further information on research design is available in the Nature Portfolio Reporting Summary linked to this article.

## Data availability

Mass spectrometry raw files have been deposited to the ProteomeXchange Consortium via the PRIDE[105] partner repository. The accession codes for the uploaded data are as follows: PRISMA screen; PXD043787, AP-MS interactome studies; PXD046950, BioID-proximity labeling; PXD043789 and Alanine Scanning: PXD043788. The processed mass spectrometry data is provided in the supplementary files. The 14-3-3ε with the GATAD1 95-LRNTKYKpSAPAAEKK-109 peptide structure has been submitted to the Protein Data Bank under the code 8Q1S. Raw data used for the quantification plots in Fig. 6C, D is provided as source data.

## Code availability

The code used to analyze the PRISMA data is available on https://github.com/Trruste/PRISMA-phosphoarray/blob/main/PRISMA_script.Rmd[106] and the code for a more detailed SLiM analysis is available on https://github.com/BIMSBbioinfo/collab_rrustemi_selbach_prisma[107].

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

## Acknowledgements

We would like to thank Martha Hergeselle and Christian Sommer for technical support, and Tobias Bock-Bierbaum and Carola Bernert for the

expression and purification of the 14-3-3 protein. We further thank the advanced light microscopy facility at the MDC, especially Anca Margineanu for her help with microscopy and image analysis. T.RR. was partially funded by the Berlin School of Integrative Oncology (BSIO) and K.M. conducted the experiments as a JSPS International Research Fellows at Kyoto University, Japan as part of the JSPS Summer Program (ID Number: SP1831).

## Author contributions

T.RR., K.M., and M.S. contributed to the design of the experiments and implementation of the project. T.RR., K.M., and Y.R. conducted experiments. T.RR., Y.R., and B.U. processed and analyzed the experimental data. A.A., K.I., Y.I., O.D., and M.S. supervised the work and provided resources. T.RR. visualized the data and produced most of the figures. T.RR. and M.S. wrote the manuscript with contributions from all co-authors.

## Funding

## Competing interests

The authors declare no competing interests.
