## [Peer Review File · Nature Communications]

Pathogenic mutations of human phosphorylation sites affect protein-protein interactionsEditorial Note: Parts of this Peer Review File have been redacted as indicated to remove third-party material where no permission to publish could be obtained.

REVIEWER COMMENTS

Reviewer #1 (Remarks to the Author):

The manuscript by Rrustemi et al describes the analysis of how 36 disease-causing (or disease associated?) mutations of phosphosites in the intrinsically disordered regions of the proteome affect protein-protein interactions using PRISMA. The authors zoom into a particular phosphorylation-dependent interaction between a peptide from GATAD1 and 14-3-3 proteins, which they characterize from a biophysical and structural level. They further suggest that the phospho-dependent interaction regulate nuclear localization of GATAD1.

While I conceptually appreciate the study, there are a couple of things that needs to be better explained out.

- Are all mutations tested disease causing (as indicated in the abstract), or are they disease associated as indicated in fig 1? Please be careful with the vocabulary, otherwise it becomes misleading.
- The information on which (and how many) proteins the peptides are from, is only provided in the sup file. Further, maybe I missed it, but I miss a statement specifying how many of the modifications (and in how many peptides and from how many proteins) that generated differential interactomes. By starring at the network in fig 2 it is probably possible to decipher it, but it is fairly challenging.
- The authors state that they filtered for motif containing peptides, but they don't specify what motifs that were found in the peptides beyond the obvious phospho-dependent motifs (e.g. pY binding to SH2 domains). Do the peptides contain other types of motifs? If so, please mention it. For example, the DENND2A peptides seems to contain an SH3 binding motif, did it pull down any SH3 domain containing proteins? This would be helpful information to understand how many of the putative SLiM-based interactions that PRISMA returns.
- Page 7: The authors state that they provide insights into SLiM-dependent interactions and refer to fig S1C. This is a very vague statement. Please help the reader and spell out what kind of insights that you provide. It is not clear to me after reading the paper.
- About the experimental design. There seems to be relatively few interactions with the wt peptides that are affected by the mutations, and that most of the interactions that are lost upon mutations would be with the phosphorylated peptides, correct? Possibly this is a consequence of the study design, as the peptides are centered around the phosphorylated residue. The recent study by Kliche et al (PMID: 37219487) showed that phosphorylation often modulates affinities of interactions when occurring in motif flanking regions. Given the length of the peptide and the focus on the phosphosite it is likely that motif-based interactions that could have been affected by phosphorylation are missed as the motifs are not well covered by the peptides. Please comment on this in the discussion.
- About the network. All interactions found by pulling with peptides with disease mutations from different diseases are in one network. It would make more sense (and be more informative) to follow the classification given in fig 1 and have one network for types of disease (e.g. one for cancer and one for mendelian diseases).
- Looking at the interactions there seems to be some neointeractions. Indeed, it is stated in the text that

there are 32 interactions that are preferentially with mutated peptides. It would be good to elaborate further on this in the text and in the discussion as it is an interesting finding.

- Also, while it is great that the authors are conservative in the analysis using known motifs for filtering, they disregard the possibility that they may uncover novel types of phospho-peptide binding domains, which would have been quite exciting. Please comment on this in the discussion.
- The 14-3-3 part seems well performed and it is a nice touch with the structure. However, the authors should cite a couple of recent important papers on family wide assessments of 14-3-3 specificities (Gogol et al (PMID: 33723253) and Segal et al (PMID: 36931259). In these papers it was convincingly shown that 14-3-3s have similar specificities, which the current results are confirming.
- The authors spent a lot of effort demonstrating that a phosphomimetic mutation is not a good strategy for probing 14-3-3 domain interactions. This is not a new finding. It is already well established in the field and the authors should cite relevant literature.
- The (potential) competition between 14-3-3 proteins and the nuclear import machinery is neat, although it is not clear which of the proteins bind directly to the peptide. The importins are fairly big proteins and challenging to work with, so I understand why they did not try to specifically validate the proposed interactions, but it is a bit unsatisfying.
- The discussion is focused on the GATA4 case, and PRISMA as a method, and fails to bring the results to a broader context. The author should provide a well-founded discussion on what novel insights the study provides in terms of “how pathogenic mutations of human phosphorylation sites affect protein-protein interactions”, which the title implies is a main focus of the study.

I am aware that some of the information requested is available in the supfiles, but it would be good to save the reader the detective work. Thank you for letting me review this manuscript.

Reviewer #2 (Remarks to the Author):

This manuscript reports a detailed investigation of 36 disease-causing mutations that affect the phosphorylation of motifs involved in protein-protein interactions. Peptide-based interaction proteomics data revealed significant differences in interactomes, often caused by disrupted phosphorylation motifs. This part of the presented study is carefully performed and interesting and suggest a number of potentially novel protein-protein interactions that could help elucidate the impact of disease-causing mutations within short linear motifs.

Major concerns:

The second part of the manuscript, aimed at identifying a novel phosphorylation-dependent interaction between the transcription factor GATAD1 and 14-3-3 proteins, is incomplete and requires additional data to strengthen the authors' conclusions.

Although the present data do suggest that motif TKYK(pS)AP of GATAD1 could be a binding motif for 14-3-3 proteins, this cannot be taken as evidence that these proteins do indeed interact in vivo and that this site is responsible for this interaction. Such a claim should be demonstrated at the full-length protein level, e.g. by co-immunoprecipitation or pulldown experiments comparing GATAD1 WT and the S102A

mutant.

Furthermore, the interaction between GATAD1 and 14-3-3 may require an additional 14-3-3 binding site, specially due to the weak affinity of the S102-containing motif. The low binding affinity could suggest that this is only an "auxiliary" binding site, with another motif acting as a "gatekeeper" site (Yaffe et al. FEBS Letters 513 (2002) 53-57). Does the GATAD1 sequence contain any additional candidate 14-3-3 binding motifs? What does prediction using 1433pred (<https://www.compbio.dundee.ac.uk/1433pred>) say?

The data suggesting that the motif containing S102 could be NLS are compelling. However, the inhibitory effect on the function of this NLS may be due to phosphorylation alone (introduction of a negative charge) and may not be related to 14-3-3 protein binding. Again, the claim that 14-3-3 binding to the pS102-containing motif regulates the function of this NLS and thus the cellular localization of GATAD1 would require additional data showing the association of these proteins in the cytoplasm as a function of S102 phosphorylation.

Additional concern:

The authors should also show the ITC titration data (i.e., not just the final binding isotherm) in Figure 5A (or at least in the supplement).

Reviewer #3 (Remarks to the Author):

In this work, Rrustemi et al. employed the PRISMA method to study the influence of mutations and phosphorylation on protein-protein interactions. The authors started from a mutation database to select 38 peptides with unique mutations in intrinsically disordered regions and with specific phosphorylation, then prepared 117 peptide spots from the 38 peptides as well as one positive control EGFR-derived phosphopeptide, in wild type, with mutation and with phosphorylation to perform the PRISMA assay using SILAC for quantification. 170 differential specific interactions were identified using the HEK293 cell. Afterwards, the authors pay specific attention to the interaction between 14-3-3 family of proteins and a GATAD1-derived phosphopeptide. It was found that the binding is regulated by the phosphorylation on GATAD1 S102, a modification site specifically observed in the heart in vivo. Finally, the authors show that the 14-3-3 binding region of GATAD1 is a functional nuclear localization signal, suggesting that binding of 14-3-3 proteins to phosphorylated GATAD1 could affect its subcellular localization. Overall, this is a very comprehensive work demonstrating the importance of phosphorylation and mutation in protein-protein interaction. The work also demonstrated the value of the PRISMA method in the study of protein-protein interactions. As pointed out by the authors, a main limitation of the current work is the lack of efficient in vivo model with the GATAD1 pS102. However, the authors performed sufficient biological experiments in vitro and in vivo to demonstrate their conclusion. The work is well designed, with comprehensive data, and the manuscript is well written. I have only a few comments for the authors' consideration.

1. Two quantitative filters were applied to identify proteins that exhibit both specific interactions with a

particular peptide and are influenced by its phosphorylation and mutation state. Although it is reported in the previous publication, it can be briefly explained here to facilitate understanding the work.

2. The BioID experiment was performed to study the interaction of 14-3-3 family proteins with GATAD1 phosphorylated on serine, and negative results were obtained due to the lack of pS102 in the HEK293 cell. However, the experiments may support the other findings in the PRISMA assay. Do you observe the interactions between the GATAD1 and other proteins by BioID as those by PRISMA? Some results can be presented.

3. From the 170 differential specific interactions, why a focus was given to the interaction between the 14-3-3 family of proteins and the GATAD1-derived phosphopeptide? It should be better explained.

Reviewer #4 (Remarks to the Author):

In this very interesting work, Rrustemi et al. investigate the phospho-dependent peptide-protein interactions focusing on a subset of peptides harboring pathogenic mutations in intrinsically disordered regions. Through databases analyses and iterative selection based on the aforementioned criteria (plus no additional mutations flanking the phosphorylated residue), they selected disease candidate peptides to perform PRISMA analysis. Each peptide was produced on a membrane in 3 versions: wt, wt phosphorylated and mutant. Following a methodological validation on a well-known EGFR phosphosite, they ran their pipeline on the remaining 37 peptide triplets. Their approach is systematic and provides valuable information, as they identify 70/111 peptides showing at least one specific interactor and 31/37 triplets for which one variant has preferential partner(s). Expectedly and supporting their approach, bait peptides containing SH2 domain binding motifs are enriched in partners containing SH2 domains, [S/T]P containing peptides capture PIN1 and they explain the frequent binding of MMTAG2, ARL6IP4 and PC4 to negatively charged (phosphorylated) peptides through their stretch of basic residues. The authors then decided to follow up on GATAD1 since its pS102 peptide interacts with several 14-3-3 proteins. They performed BioID on 4 different constructs: GATAD1 WT, S102A (phosphodead), S102D (phosphomimetic) and S102P (natural mutant). Unexpectedly, the WT and S102D mutant did not detect the 14-3-3 proteins as proximal interactors. The authors then performed ITC between the different GATAD1 peptides and YWHAE, which convincingly shows that the phosphomimetic mutant do not recapitulate the ability of the phosphopeptide to interact with GATAD1. Furthermore, through alanine (glycine for alanine) scanning, they identified additional residues (+1A and +2P) critical in the epsilon 14-3-3 interaction. The structural analysis revealed that none of the non-phosphorylated variant could form an interaction with YWHAE given the distance between the residue 102 and the positively charged triad patch of YWHA. To gain insight into the functional significance of the pS102 peptide-14-3-3 interactions, the authors finally performed a PRISMA analysis with all the alanine scanning peptides + the variant pS102S/A/D/P, which they prove were unable to bind the 14-3-3 proteins. Strikingly, they identified an enrichment in nuclear import proteins suggesting that the binding of 14-3-3 proteins on the pS102 peptides could impair the nuclear import of GATAD1. Using the NLStradamus model, they identified a region spanning the 14-3-3 interacting peptide as a candidate NLS, which they confirming as a bona fide NLS fusing it to GFP. They

finally conclude that the 14-3-3 could mask the pS102 GATAD1 NLS, impairing its proper transport into the nucleus.

It is notoriously difficult to study PTM-dependent interactions. I find the approach of the authors interesting and I acknowledge their application to present both positive (PRISMA, alanine scanning and structural analysis) and negative data (BioID). Their work appears conceptually important to me. For example, the ITC experiments are clearly showing that phosphomimetic peptides are not recapitulating all the features of a true phosphorylation, structurally well supported by the lack of proximity between the negative charge and the modified residue. Overall, I believe this work is of high quality, despite a few limitations that the authors are aware of (e.g. lack of model system) and a few reserves listed below that I think should be lifted to reach the publication level of Nature Communications and make this paper more impactful.

In summary, the pathophysiological mechanism of this mutation could come from the defect of cytoplasmic sequestration of GATAD1 in cardiomyocytes, from the expression of a S102P variant in the nucleus or from the absence of the GATAD1wt. There is no insight about what mechanism is at play. However, even if the biological conclusion of this paper falls short and that interactomic study needs to be reworked, the methodological approaches and the results are of high quality. The follow up on GATAD1-14-3-3 interaction is complicated and the authors proposed a convincing mechanism that could be supported by a few additional assays. I would thus consider this paper suitable for Nature Communications following the revisions listed below.

Major comments:

The BioID part of the manuscript is puzzling me. I find it laudable that the authors tried to investigate the relevance of the GATAD1 peptide variants using whole protein variants expressed in living cells and that they include these data to the manuscript, even if they are not really supportive of their PRISMA findings. Given the results obtained in the rest of the manuscript, the data are not very surprising. However, I have been through the BioID supplemental table, which contains very few information, then looked at the proteingroup file deposited on the proteomeXchange server.

- I have been trying to understand where did the ratio presented in the supplemental table come from and was unable to find the way that the authors used to get these results. Please include additional tabs with the data used to calculate these ratios and explain how you calculated them.

- The common way of doing BioID is filtering out the background using the BirA* tag alone, not the 'no biotin'. I do not understand how your data could define the basal proximal interactome of GATAD1 comparing GATAD1wt to 'no biotin'.

- I do not understand how you pass from 3,052 protein groups identified across the conditions to 183 (actually 152 because 31 are NA for all baits according to the supplemental table) protein groups in the table using the 'no biotin' condition as a sole filtering condition.

- Performing SILAC for BioID is not common, but I acknowledge that it could help multiplexing and maybe reducing the MS analysis cost. This is at the expense of analysis depth, but it is not the main issue in interactomics studies.

- The only valuable data of the BioID part are the comparison between the mutants and the WT, but again, I would need additional information to understand where the ratios came from.

- From my understanding, the label swap means that in one sample, WT is heavy and the other condition

is light isotope, and in the other, the WT is light and the other condition is heavy. I was not able to identify what column corresponded to what condition.

Overall, I have been struggling a lot to get a feel about these data. Although from the presented results, it seems that the variants do not change much the interactome (which sometimes happens), I am unable to validate the results in their current form. At the very least, the authors should present their data in a comprehensive manner, using additional tabs to show how they arrived to their hit list from the MaxQuant output (there are no statistical values either). I have been sincerely trying to find the authors' way using the MaxQuant output but unsuccessfully. In addition, they could either remove the part on the basal interactome of GATAD1 WT or do it as suggested below, because comparing the data between the bait and 'no biotin' is not the right way to do it. The comparison between the WT and mutants is what really matters here, and even if there are only duplicates (acceptable for MS data), the important result is that they see no real differences (I would like to see the calculation process to get to this conclusion though). If they choose to do these corrections and not perform additional BioID experiments, I would suggest putting this part of the results in supplemental data and presenting it as a pilot experiment (since the main result is that the mutants do not follow the PRISMA interaction pattern with the 14-3-3 proteins, and that they further show that it is due to (i) an imperfect mimicking of phosphorylation by the S102D; (ii) a transport of all constructs in the nucleus).

To make the experiments done in cells insightful, I would suggest two different options:

1. If the authors really want to keep the BioID data in the core of their paper, I think they could quickly fix the aforementioned issues since they have made the stable cell lines. They can redo the whole BioID part, using regular culture conditions (not SILAC) and analyze independent triplicates for each cell line, adding a triplicate of BirA* alone samples as controls (that would be 15 samples). Alternatively, for the controls, it would also be acceptable to use files from the CRAPome depository (Mellacheruvu et al., Nat Methods, 2013; <https://reprint-apms.org/>) to remove background (there are BirA* alone files in HEK293 Flp-In TREx cells in there, but make sure they have been analyzed on an Exploris). The comparison of BirA*-GATAD1 with BirA* alone would provide the real basal proximal interactome of GATAD1. The 'no biotin' or 'no tet' controls are not necessary since the BirA* alone samples will be more stringent. Just use 50µM biotin everywhere.

2. If they opt for moving the BioID data in supplemental, I think they should show additional supportive evidence that the GATAD1 NLS they propose is functional, e.g. through transiently transfecting a candidate NLS mutant into HEK293 cells along with the other variants they used for BioID. Also, given that the S102E variant was structurally better at mimicking the phosphosite, I would add it in this experiment (or ideally as an additional BioID bait).

Suggestions: This is a little beyond the main message of the paper, so, unless the authors have already done some of the following experiments (even if the data are negative), I do not expect them to be included in a revised version. All the experiments in cellular systems could be performed in GATAD1wt depleted cells. According to Theis et al., the S102P mutation is pathogenic when homozygote in patients. This suggests that performing a set of experiments in a GATAD1 KD or KO background could be the best way to gather biologically relevant data in cellular systems. BioID could be more efficient using the TurboID tag (Branon et al., Nat Biotech, 2018; ~30 min labeling time instead of 18 h for regular BioID), in case the residence time of GATAD1 is too short for sufficient labeling in the cytoplasm. In these settings,

it might be worth performing the analysis on cytoplasmic and nuclear fractions.

Finally, there is the common issue of in vitro binding analyses between proteins that are localized in different subcellular compartments. Biochemically possible interaction could be biologically rare or impossible. I think this is the case for GATAD1 (nuclear) and the 14-3-3 proteins (cytoplasmic). As the authors state, the model system is lacking and they cannot easily investigate this interaction in living cells. It could indeed be specific of the cardiomyocytes since this mutation is linked to dilated cardiomyopathy, but the biological evidences are lacking. The hypothesis that the 14-3-3 proteins could mask the NLS of GATAD1 suggests that a cardiomyocyte-specific kinase could phosphorylate this residue. From the PRISMA data, the authors' choice is obvious but the discrepancy between the basal localization of the 14-3-3 proteins and GATAD1 render the functional studies much more complicated. This is why one of the two options proposed above appear necessary to support their results.

Minor comments:

- Page 32: 50 μ M biotin instead of 50mM
- The streptavidin-sepharose beads from GE vary from lot to lot, sometimes they are coated with too many streptavidin which generates too many streptavidin peptides following on beads digestion (Saint-Germain et al., J. Proteome Res., 2020). I checked the MS raw files and they are fine. For future studies, I would recommend to use the Pierce Ultralink beads instead.
- The Tomioka et al., reference is incomplete (BioRxiv; use the citation tool to import the ref)
- For the PRISMA analyses, I did not find the information of the HEK293 lysis buffer and how the amounts of protein from each labeled sample (H, M, L) were normalized.
- The supplemental table show minimal information. It would be better to put the raw output of MaxQuant, then an intermediary table showing the columns useful for calculations, then the hit table. This is pretty much impossible in the current form to understand for a non-MS specialist.
- Page 12, it is stated that "All of these peptides contain an [pS/T]P motif known to interact with the WW domain of PIN1 (Lee et al, 2011)". However, in Figure 3C, 2 peptides do not interact with this motif. If correct, change to "Eight out of ten of these peptides..."
- Page 12: "MMTAG2, ARL6IP4 and PC4 interact with many phosphopeptides (9, 7 and 6, respectively)". I count 5 phosphopeptides for PC4. Maybe 2 edges are superimposed? Please check.
- Unless I missed them, please add the supplemental tables legends under the form of a "readme" tab
- Why did you choose the YWHAE and not another 14-3-3 protein?

Reviewer #1 (Remarks to the Author):

The manuscript by Rrustemi et al describes the analysis of how 36 disease-causing (or disease associated?) mutations of phosphosites in the intrinsically disordered regions of the proteome affect protein-protein interactions using PRISMA. The authors zoom into a particular phosphorylation-dependent interaction between a peptide from GATAD1 and 14-3-3 proteins, which they characterize from a biophysical and structural level. They further suggest that the phospho-dependent interaction regulate nuclear localization of GATAD1.

While I conceptually appreciate the study, there are a couple of things that needs to be better explained out.

We thank the reviewer for her/his supportive comment!

- Are all mutations tested disease causing (as indicated in the abstract), or are they disease associated as indicated in fig 1? Please be careful with the vocabulary, otherwise it becomes misleading.

This is a relevant question. The disease candidates were all selected from the PTMVar dataset in PhosphoSitePlus. This dataset contains data from modified sites that are either at or within five residues from the site that has been mutated. It contains data from the humsavar.txt from UniProtKB, from TCGA, COSMIC, cBIO and other sources. In this dataset the mutations are categorized in three categories: Unclassified, Polymorphism and Disease. In case of Mendelian disorders in humsavar they are classified as disease-causing in OMIM. For cancer-associated mutations the situation is a bit more complicated because an individual point mutation alone typically does not “cause” cancer. To be classified as “disease” in PTMvar, somatic mutations from cancer genomes are included only when observed “*in three to five independent cancer patient samples.*” (Hornbeck et al. 2015)

To make disease classification for the variants more clear, we inserted the following sentences into the Material and Methods section:

“In the database, mutations are categorized as polymorphism, unclassified, or disease. Mutations labeled as ‘disease’ are those associated with Mendelian diseases, while somatic cancer mutations are designated as ‘disease’ only if observed in three to five distinct cancer patients”

In addition, we inserted a reference to the Material and Methods part in the Results section:

“see Material and Methods for more details on variant classification.”

In addition, we changed the word “disease-causing” to “disease-associated” throughout the manuscript since this is the more generic term.

- The information on which (and how many) proteins the peptides are from, is only provided in the sup file. Further, maybe I missed it, but I miss a statement specifying how many of the modifications (and in how many peptides and from how many proteins) that generated differential interactomes. By starring at the network in fig 2 it is probably possible to decipher it, but it is fairly challenging.

This point is well taken. In general, it is difficult to decide how much information should be presented in the main text or in the supplement. To answer the question: The 38 candidates we selected cover 34 different proteins. We added this information to the Results section:

“This filtering resulted in a final set of 38 disease candidates derived from 34 different proteins (Figure 1A, Supplemental Table 1).”

The data requested by the reviewer is contained in supplemental table 1. This table is rather long, which is why we decided to keep it in the supplement.

Regarding the second part of your question about the number of peptides (based on modification) that produced differential interactomes, Figure 2 aims to address this query.

In Panel D of Figure 2, a pie chart illustrates that out of 111 total peptides, 70 peptides specifically interacted with at least one protein. Moreover, out of 37 peptide triplets, 31 displayed at least one unique and differential interactor.

For a more detailed view, we've expanded on these findings in Figure S3 panel A. The pie charts (middle lane) in this figure shows that out of the 70 peptides with specific interactors, 28 are phosphorylated, 19 are mutated, and 23 are wild-type. Delving deeper, the pie charts on the right display the number of peptide forms with at least one specific and differential interactor. Of the 28 phospho-peptides with specific interactions, 26 also have differential ones. In contrast, 10 of the 19 mutated peptides and 9 of the 23 wild-type peptides present differential interactions.

- The authors state that they filtered for motif containing peptides, but they don't specify what motifs that were found in the peptides beyond the obvious phospho-dependent motifs (e.g. pY binding to SH2 domains). Do the peptides contain other types of motifs? If so, please mention it. For example, the DENND2A peptides seems to contain an SH3 binding motif, did it pull down any SH3 domain containing proteins? This would be helpful information to understand how many of the putative SLiM-based interactions that PRISMA returns.

We apparently did not make this clear enough. The information about all motifs (in the peptides) and cognate domains in interaction partners (in proteins identified in the pull-downs) is contained in Table S3. We now added information to this table to clarify what the individual columns mean and hope this clarifies what exactly the table shows. Importantly, while this table shows all annotated SLiM-domain pairs (for the peptides we used and the proteins we identified in the pulldowns), this does not prove a physical interaction between the peptide and the protein. In the end, ELM is just a motif database – establishing true physical interactions requires experiments such as the peptide pulldown we performed.

About DENND2A: This peptide indeed harbors a PxxP motif annotated to interact with the SH3 domain of GRB2 (Uniprot ID P62993) and SPTAN1 (Uniprot ID Q13813). In both cases, the motif is found in both the wild-type and the mutated form of the peptide. This information can be found in supplementary table 3. Looking at the volcano plots for DENND2A shows that GRB2

indeed appears to be a specific binder of the wild-type and the mutated peptide. However, it did not pass our rather stringent cut-offs (see volcano plots below). In contrast, SPTAN1 does not appear to bind specifically. Hence, although ELM predicts a SPTAN1-interacting SLiM in this peptide, the motif does not actually appear to mediate interaction. This highlight again the importance of complementing predictions with experimental data.

Figure R1: Volcano plots for DENND2A-derived peptides with GRB2 and SPTAN1 highlighted. The volcano plots are the same as in figure S3

- Page 7: The authors state that they provide insights into SLiM-dependent interactions and refer to fig S1C. This is a very vague statement. Please help the reader and spell out what kind of insights that you provide. It is not clear to me after reading the paper.

This comment also indicates that we did not provide sufficient information about how we carried out the analysis of LFQ score distributions in peptide-protein interactions in the context of SLiM-Domain interactions.

The primary mode of action of SLiMs is through binding to (usually) a larger globular and structured protein domain. These SLiM-Domain class pairs have been annotated in the ELM database (also see Kumar et al. 2020). Based on this paradigm, observing known SLiM patterns in a given peptide in our screen and a cognate PFAM domain in its detected interaction partners provides us an additional layer of confidence in the detected interactions. In our screen, we typically identified >500 proteins in individual pull-downs, most of them probably unspecific. To assess if our data reflects known SLiM-domain relationships we took advantage of LFQ values since they allow us to compare abundances of proteins across the different pull-downs. The rationale is that whenever an interaction can be explained by a SLiM-domain pair (that is, whenever a protein identified in a given pull-down has a PFAM domain matching to a SLiM observed in the sequence of the respective peptide), the LFQ value of the protein should be higher than in other pull-downs (peptides w/o a SLiM matching to the PFAM domain). Thus, observing that interactions that can be explained by one or more SLiM-Domain pairs have

higher LFQ score distributions compared to those that cannot be explained with any known SLiM-Domain pairs serves as supporting evidence for the biological relevance of the results obtained from the screen. For this analysis, we considered all known ELM classes (356 classes) and all known PFAM domains (180 unique PFAM domains) that are known to recognize these SLiMs.

The plot in figure S1C addresses the above question globally for all possible SLiMs and PFAM domains. We now included a more detailed analysis for individual domain-motif pairs to provide more clarity, as requested by the reviewer. The new figure S2 compares protein abundances (that is, z-scored LFQ values) broken down by the top SLiM-domains. This compares LFQ scores of proteins that include a domain (e.g. SH3) w.r.t whether those proteins are ever paired by the peptide via a cognate slim (e.g. LIG_SH3) (also grouped into by peptide type; phos/wt/mut). For example, we observe that proteins containing a WW domain preferentially bind to peptides harboring corresponding motifs. Importantly, 14-3-3, SH2 and WW domain-containing proteins bind preferentially to the phosphorylated form of the peptide, consistent with the phosphorylation-dependent nature of these interactions. In contrast, SH3 domain-containing proteins interact with cognate SLiMs in a phosphorylation independent manner. Overall, these data show that the observed global relationship also holds for individual SLiM-domain pairs.

We added the figure below as a new supplemental figure S2 to the manuscript. Also, we added the following text to the paper:

“To provide a more detailed picture, we looked at the four most frequently observed SLiMs mediating interactions (motif class type “DOC” and “LIG” in ELM) (Figure S2). For example, we observe that proteins containing a WW domain preferentially bind to peptides harboring corresponding motifs. Importantly, 14-3-3, SH2 and WW domain-containing proteins bind preferentially to the phosphorylated form of the peptide, consistent with the phosphorylation-dependent nature of these interactions. In contrast, SH3 domain-containing proteins interact with cognate SLiMs in a phosphorylation independent manner (Figure S2). Overall, these data show that the observed global relationship also holds for individual SLiM-domain pairs.”

We have documented the code, figures, and the methods for this analysis on the following github repository: https://github.com/BIMSBbioinfo/collab_rrustemi_selbach_prisma

- About the experimental design. There seems to be relatively few interactions with the wt peptides that are affected by the mutations, and that most of the interactions that are lost upon mutations would be with the phosphorylated peptides, correct? Possibly this is a consequence of the study design, as the peptides are centered around the phosphorylated residue. The recent study by Kliche et al (PMID: 37219487) showed that phosphorylation often modulates affinities of interactions when occurring in motif flanking regions. Given the length of the peptide

and the focus on the phosphosite it is likely that motif-based interactions that could have been affected by phosphorylation are missed as the motifs are not well covered by the peptides. Please comment on this in the discussion.

We fully agree: Our study is indeed focused on mutations affecting the phosphorylated residue itself. Therefore, we do not cover cases when a mutation occurs in a flanking region of a phosphorylation site. This may explain why we indeed find more differential interactions for the phosphorylated form (n = 105) than for the wt (n=33) and mutant form (n=32). This information is displayed in Figure 2 E and supplemental figure S3B.

We also fully agree that our results are impacted by the study design. Also, we believe that extending the work to mutations adjacent to phosphorylation sites would be very interesting. In fact, we already discussed this point in the discussion:

“In the future, it will be interesting to also investigate mutations adjacent to phosphorylation sites, not just changes of the phosphorylated residue itself. Such mutations can also modify SLiMs and thereby change PPIs. For example, the lung cancer associated P1019L mutation in EGFR has already been shown to switch the binding specificity of the adjacent phosphorylation site pY1016 (Lundby et al, 2019).”

We now extended this by also referring to the paper mentioned by the reviewer:

“Also, a recent study showed that phosphorylation often modulates affinities of interactions when occurring in motif flanking regions (Kliche et al. 2023).”

- About the network. All interactions found by pulling with peptides with disease mutations from different diseases are in one network. It would make more sense (and be more informative) to follow the classification given in fig 1 and have one network for types of disease (e.g. one for cancer and one for mendelian diseases).

Displaying separate networks for different disease types would be an option. However, even within one disease type (OMIM, COSMIC, TCGA) there are several different diseases affecting different organs. So one could even argue that we should display individual networks for each disease, which would result in many separate networks. Instead, we chose to display all interactions in a single network because this highlights interactions shared by multiple peptides across disease entities. For example, as mentioned in the manuscript, we see that Pin1 interacts with several peptides containing a WW-domain binding motif. This information would be lost if we split the network. Similar cases can be made for MMTAG, PC4, and ARL6IP. Therefore, we decided to keep the network as is. We show separate cancer and OMIM subnetworks below for the reviewer.

Peptide-protein network separated by disease

TCGA and COSMIC mutation candidates

At least of the peptide forms of 16 out of 20 candidates associated with cancer showed one or more specific and differential interactors

OMIM database mutation candidates

At least one of the peptide forms of 14 out of 16 candidates that cause Mendelian diseases showed one or more specific and differential interactors

Figure R2. Separated peptide-protein network according to the disease type. Top Network: Disease candidates implicated with cancer development. These candidates were selected from PTMVar database but originally are sourced from either TCGA or COSMIC databases. From 20 mutation candidates 16 show at least one specific or differential interactor (Top pie chart). These interactions could be differential for any peptide form.

Bottom Network: Disease candidates that represent Mendelian Disease. These mutations are deposited in the PTMVar dataset from the OMIM database. 16 Mendelian disease causing mutations were studied on the screen. From these 14 have at least one specific and differential interactor (Bottom pie chart).

- Looking at the interactions there seems to be some neointeractions. Indeed, it is stated in the text that there are 32 interactions that are preferentially with mutated peptides. It would be good to elaborate further on this in the text and in the discussion as it is an interesting finding.

This is an interesting point. Indeed, there are 32 interactions that are differential for the mutated form (Fig. 2 E), of which 28 are unique for the mutated form (Fig. S2B). This shows that mutations cannot only result in a loss of interaction but also cause new ones. We have in fact observed this previously in a study that did not investigate the role of phosphorylation. In this previous work, we found that mutations can create novel dileucine motifs that are pathogenic because they can cause clathrin-dependent endocytosis of transmembrane proteins (Meyer et al., Cell, 2018).

To highlight this point we added the following paragraph to the discussion section:

“We observed 58 interactions that are differential between the wild-type and the mutant (30 lost and 28 gained upon mutation). In this context, it is important to highlight that our screen can identify interactions that are gained upon mutation – an aspect often overlooked when investigating disease mutations (Fig. S2B). We previously reported that pathogenic mutations can cause disease by creating dileucine motifs that lead to clathrin-binding (Meyer et al. 2018). It would be interesting to follow up on the potential novel interactions of the mutated forms identified here ”

- Also, while it is great that the authors are conservative in the analysis using known motifs for filtering, they disregard the possibility that they may uncover novel types of phospho-peptide binding domains, which would have been quite exciting. Please comment on this in the discussion.

This is a good point. However, we feel that the 38 sites we have in this work are too few to really identify novel types of phospho-peptide binding domains/motifs. In the light of the increased throughput of proteomics, this is an exciting future application of the PRISMA technology. To highlight this we added the following sentence to the discussion:

“Furthermore, exploring a broader spectrum of sites and incorporating various types of modifications could reveal previously unknown domain-motif relationships.”

- The 14-3-3 part seems well performed and it is a nice touch with the structure. However, the authors should cite a couple of recent important papers on family wide assessments of 14-3-3 specificities (Gogol et al (PMID: 33723253) and Segal et al (PMID: 36931259). In these papers it was convincingly shown that 14-3-3s have similar specificities, which the current results are confirming.

Thanks for bringing these papers to our attention! We updated the text in the results section from:

“First, all six 14-3-3 family proteins show essentially identical binding preferences. This is interesting since – despite the relatively high sequence homology – not all ligands show equal affinity for different 14-3-3 isoforms (Gardino et al, 2006).”

to:

“First, all six 14-3-3 family proteins show essentially identical binding preferences. This observation is in line with recent studies indicating that 14-3-3 paralogs have very similar target-binding tendencies (Segal et al. 2023; Gogl et al. 2021)”

- The authors spent a lot of effort demonstrating that a phosphomimetic mutation is not a good strategy for probing 14-3-3 domain interactions. This is not a new finding. It is already well established in the field and the authors should cite relevant literature.

We thank the reviewer for pointing this out. We fully agree that this is not a new finding, and we did in fact already cite the paper by Kozelekova et al, 2022 that shows this especially for 14-3-3 proteins but did not explain this in detail. We now included additional references for 14-3-3 proteins and provide more detailed information in the text:

*“It should be noted that phosphomimetic mutations do not recapitulate all features of a phosphorylated residue (Dephore et al, 2013; Pérez-Mejías et al, 2020). Although, phosphomimetic mutations have been successfully employed to imitate phosphorylation in many cases (Vieira-Vieira et al, 2022; Thorsness & Koshland, 1987; Koyano et al, 2014; Imami et al, 2018), **there are instances where they do not effectively replace phosphorylated amino acids. This is especially relevant for interactions to 14-3-3 proteins where phosphomimetic mutations mostly failed to mimic phosphorylation** (Vander Haar et al. 2007; Johnson et al. 2010; Courchet et al. 2008; Suhda et al. 2023; Kozeleková et al. 2022; Gogl et al. 2020), except for one example (Faul et al. 2007).”*

- The (potential) competition between 14-3-3 proteins and the nuclear import machinery is neat, although it is not clear which of the proteins bind directly to the peptide. The importins are fairly big proteins and challenging to work with, so I understand why they did not try to specifically validate the proposed interactions, but it is a bit unsatisfying.

We fully agree with this comment. In fact, we tried long and hard to set-up a cellular model system to assess how the interaction of 14-3-3s, GATAD1 and importins affects the subcellular localisation of GATAD1 in a phosphorylation dependent manner. Unfortunately, due to the inability of the phosphomimetic mutant to actually mimic phosphorylation, we were not able to do this.

We could indeed try to biochemically assess the interaction of GATAD1 with importins. However, as the reviewer points out, it is quite challenging to work with them since they are big. Also, we see several importins in our proteomic data (IPO4, IPO5, IPO7, TNPO3) and they have a poorly defined binding specificity, which makes it difficult to choose candidate for ITC experiments. Most importantly, even if we could do these experiments, this would not prove the phosphorylation and 14-3-3 dependent regulation of nuclear import of GATAD1.

Therefore, we decided not to perform ITC experiments with GATAD1 and importins. Instead, we investigated the function of the GATAD1 NLS in more detail. In the original manuscript we did already show that attaching the potential GATAD1 NLS to GFP induces nuclear import. Now, we also performed a loss of function experiment (new figure 6D): We deleted the NLS region of GATAD1 and assessed the impact of this deletion on the subcellular localisation of the protein. We observed that GATAD1 lacking the NLS region is clearly more cytoplasmic than the wild-type protein. In combination with the GFP experiment, these data show that the sequence is both required and sufficient for nuclear localisation of GATAD1. Hence, we can now say that this region constitutes a functional NLS.

- The discussion is focused on the GATA4 case, and PRISMA as a method, and fails to bring the results to a broader context. The author should provide a well-founded discussion on what novel insights the study provides in terms of “how pathogenic mutations of human phosphorylation sites affect protein-protein interactions”, which the title implies is a main focus of the study.

I am aware that some of the information requested is available in the supfiles, but it would be good to save the reader the detective work. Thank you for letting me review this manuscript.

This point is again well taken. We changed the discussion to highlight conclusions we can draw at a more global level. The corresponding section is:

“Although proteomics can now routinely identify thousands of phosphorylation sites, their functional characterisation is lagging behind (Needham et al. 2019). Recently, a number of computational (Ochoa et al. 2020; Bludau et al. 2022) and experimental approaches (Vieira-Vieira et al. 2022; J. M. Lee et al. 2023; Imami et al. 2018; J. L. Johnson et al. 2023) have been developed to assess the functional relevance of phosphopeptides on a more global scale. Here, we took advantage of the established PRISMA method (Ramberger, Sapozhnikova, et al. 2021; Kassa et al. 2023; Meyer et al. 2018) to study the function of phosphorylation sites coinciding with disease-associated mutations. In principle, phosphorylation can both induce and disrupt interactions. Of the 170 specific and differential peptide-protein interactions we

observed, 132 were affected by the peptide phosphorylation state (see Fig. 2). Interestingly, we observed 102 phosphorylation-induced interactions while only 30 were disrupted. Hence, phosphorylation mostly tends to induce interactions. Importantly, almost all of the phosphorylation-induced interactions (101 out of 102) were disrupted by the mutations, indicating that their loss could indeed cause disease. While we observed more phosphorylation-dependent interactions in our screen, it is important to point out that mutations could also affect interactions independently of the phosphorylation state. We observed 58 interactions that are differential between the wild-type and the mutant (30 lost and 28 gained upon mutation). In this context, it is important to highlight that our screen can identify interactions that are gained upon mutation – an aspect often overlooked when investigating disease mutations (Fig. S2B). We previously reported that pathogenic mutations can cause disease by creating dileucine motifs that lead to clathrin-binding (Meyer et al. 2018). It would be interesting to follow up on the potential novel interactions of the mutated forms identified here. These numbers indicate that the modification state of a site has a broad impact on the interactome and the mutation state can highly disrupt these interactomes. Indicating that many of these mutations are pathogenic because they impair phosphorylation-dependent interactions. However, we do not know if this observation from our limited list of phosphorylation sites can be generalized. It is important to point out that not all observed differential interactions (phosphorylation- or mutation-affected) are necessarily disease-relevant. While we focused on one interaction here, our data potentially contains additional candidates that would be interesting to follow up on.”

Reviewer #2 (Remarks to the Author):

This manuscript reports a detailed investigation of 36 disease-causing mutations that affect the phosphorylation of motifs involved in protein-protein interactions. Peptide-based interaction proteomics data revealed significant differences in interactomes, often caused by disrupted phosphorylation motifs. This part of the presented study is carefully performed and interesting and suggest a number of potentially novel protein-protein interactions that could help elucidate the impact of disease-causing mutations within short linear motifs.

Major concerns:

The second part of the manuscript, aimed at identifying a novel phosphorylation-dependent interaction between the transcription factor GATAD1 and 14-3-3 proteins, is incomplete and requires additional data to strengthen the authors' conclusions.

Although the present data do suggest that motif TKYK(pS)AP of GATAD1 could be a binding motif for 14-3-3 proteins, this cannot be taken as evidence that these proteins do indeed interact in vivo and that this site is responsible for this interaction. Such a claim should be demonstrated at the full-length protein level, e.g. by co-immunoprecipitation or pulldown experiments comparing GATAD1 WT and the S102A mutant.

This point is well taken. However, as pointed out in the original manuscript, we were not fully able to functionally evaluate the interaction because it is phosphorylation-dependent and the phosphomimetic (S102D) mutant does not mimic the phosphorylated state. We have shown this with both ITC and X-ray crystallography experiments. Also, we would also like to point out that Reviewer #3 said that we “*performed sufficient biological experiments in vitro and in vivo to demonstrate [our] conclusion*”.

Nevertheless, we followed the suggestion of the reviewer and performed co-immunoprecipitation experiments combined with mass spectrometry (AP-MS). More specifically, we used HEK293 cells stably expressing FLAG-tagged GATAD1 wt, S102P (pathogenic variant), S102D (phosphomimetic variant) or S102A (non-phosphorylatable variant). We then asked which proteins are pulled-down specifically with each of these GATAD1 variants compared to the control (that is, non-induced control cells). These data are shown in the new Figure 4B.

We observed that several well-known GATAD1 binders interacted specifically with FLAG-tagged full-length GATAD1 such as the transcriptional regulators KDM5A, RBBP7/4 PHF12 and SIN3B. Together, these proteins form the EMSY complex, which binds to H3K4me3-marked, active promoters (Vermeulen et al, 2010; Varier et al, 2016). More globally, gene ontology enrichment analysis of the 47 proteins binding significantly to wild-type GATAD1 revealed their involvement in Sin3 complex and transcriptional corepressor activity (not shown), consistent with the known biology of GATAD1. However, we did not observe interaction of wild-type GATAD1 with 14-3-3 proteins. Consistently, the interaction partners of the non-phosphorylatable S102A mutant were very similar to the wild-type. This suggests that GATAD1 may not be phosphorylated on S102 in HEK cells. Indeed, while GATAD1 is widely expressed and multiple phosphorylation sites (T34, T52, S55, S194, S235, Y248) have been identified in different cell lines (HeLa, KG1, K562, MKN-45), the only evidence for S102 phosphorylation comes from murine heart tissue (Lundby et al, 2013; Hornbeck et al, 2015). Consistent with the ITC and X-ray crystallography data we provided (Fig. 5), the phosphomimetic S102D mutant also did not bind to 14-3-3 proteins in the new AP-MS data (Fig. 4).

The new AP-MS experiments requested by the reviewer are consistent with the BioID data we had in the original submission. We now show the new AP-MS data in the main text (Figure 4B) and move the BioID part into the supplement (Figure S6).

The new AP-MS data still does not provide evidence that GATAD1 and 14-3-3 proteins interact in vivo and that this site is responsible for this interaction. We think this is due to two factors: First, GATAD1 S102 appears to be specifically phosphorylated only in cardiomyocytes. Second, the GATAD1 S102D mutant does not mimic GATAD1 phosphorylation and does not interact with 14-3-3 proteins. As pointed out in our paper, we therefore used biochemical experiments with the phosphorylated GATAD1 peptide for follow-up studies (Fig. 5 and 6). This provided evidence for a role of S102 phosphorylation and 14-3-3 binding in regulating nuclear cytosolic trafficking of GATAD1. However, due to the lack of a suitable cellular model, this evidence is indirect. For this reason, we were very careful with our conclusions.

We write in the discussion:

“Due to the lack of a suitable model system, we cannot fully investigate the cellular consequences of the GATAD1 14-3-3 interaction.”

“Nevertheless, due to the lack of a cellular model system, we are not able to experimentally validate this hypothesis.”

Also, in the abstract we write:

*“Follow-up experiments revealed the structural basis of this interaction and **suggest** that 14-3-3 binding affects GATAD1 nucleocytoplasmic transport by masking a nuclear localisation signal.”*

We would like to stress that we never claimed that “these proteins do indeed interact in vivo and that this site is responsible for this interaction”. While we would have loved to make this claim, as both the reviewer and we pointed out, we were not able to experimentally validate this. Nevertheless, we feel that the results of our screen and the follow-up on GATAD1 provides interesting new insights. For example, we think the identification of the nuclear localisation signal of GATAD1 is relevant even without the context of 14-3-3 binding. We had already shown in the original manuscript that fusing a peptide harboring the putative NLS to GFP drives GFP to the nucleus. We now performed an additional loss of function experiment: We deleted the putative NLS from GATAD1 and investigated its subcellular distribution (new figure 6D). We find that the deletion results in a redistribution of GATAD1 to the cytosol. Hence, we can now say that the putative NLS is indeed functional since it is both required and sufficient for nuclear localisation of GATAD1.

Furthermore, the interaction between GATAD1 and 14-3-3 may require an additional 14-3-3 binding site, specially due to the weak affinity of the S102-containing motif. The low binding affinity could suggest that this is only an "auxiliary" binding site, with another motif acting as a "gatekeeper" site (Yaffe et al. FEBS Letters 513 (2002) 53-57). Does the GATAD1 sequence contain any additional candidate 14-3-3 binding motifs? What does prediction using 1433pred (<https://www.compbio.dundee.ac.uk/1433pred>) say?

This is a relevant question. No region in the GATAD1 protein contains the canonical Mode I or Mode II motifs for 14-3-3 binding. However, both Scansite and 14-3-3pred predict two and three potential 14-3-3 binding sites, respectively. In both these prediction tools, the GATAD1 peptide we study is scored the highest. Especially in the 14-3-3pred tool is the only one that passes the consensus cut off (= 0.5). Hence, according to the predictions, the S102 site we studied here appears to be the most “relevant” one. This being said, since we haven’t experimentally validated any of the other two predicted binding sites, auxiliary binding effects cannot be ruled out.

To highlight the predicted binding regions we have now added an extra figure to panel B in figure 5 of the main manuscript. We have also added the following sentence in the main text under the Structural analysis of the GATAD1 14-3-3 interaction section:

“Additionally, motif prediction platforms like Scansite (Obenauer et al. 2003) and 14-3-3pred (Madeira et al. 2015) identify this region as having the top scores for potential 14-3-3 binding, with values of 0.358 and 0.778, respectively.”

The data suggesting that the motif containing S102 could be NLS are compelling. However, the inhibitory effect on the function of this NLS may be due to phosphorylation alone (introduction of a negative charge) and may not be related to 14-3-3 protein binding. Again, the claim that 14-3-3 binding to the pS102-containing motif regulates the function of this NLS and thus the cellular localization of GATAD1 would require additional data showing the association of these proteins in the cytoplasm as a function of S102 phosphorylation.

We totally agree with the statement, but as highlighted in the manuscript, validating the interaction dynamics between GATAD1, 14-3-3s, and importins is challenging without a cellular model where GATAD1-S102 is phosphorylated or a phosphomimetic mutant that faithfully mimics phosphorylation.

Having said this, our alanine scanning experiments did include the S102D mutant peptide, despite its lack of binding to the 14-3-3 epsilon in our ITC experiment. While the S102D mutant peptide carries a negative charge, it still did not interact with any of the 14-3-3 proteins in the alanine scan. However, this peptide showed a similar interaction with importins as the other non-phosphorylated peptides (refer to Figure 6, far-right peptide). This suggests that a mere charge effect isn't sufficient to abolish the interaction between the GATAD1 peptide and importins.

To investigate this point in more detail, we now also included a more detailed analysis of GATAD1 mutations on the subcellular localisation of the protein (new figure S5). These data show that all GATAD1 point mutations localise to the nucleus, similar to the wild-type protein. This is also true for the S102D mutant. We can therefore conclude that the point mutations themselves do not appear to interfere significantly with nuclear cytosolic trafficking.

Having said all of this, we still agree with the reviewer that we cannot fully validate the hypothesis that 14-3-3 binding to GATAD1 affects nuclear cytosolic trafficking. Therefore, we were particularly careful with our conclusions.

Additional concern:

The authors should also show the ITC titration data (i.e., not just the final binding isotherm) in Figure 5A (or at least in the supplement).

We thank the reviewer for pointing this out. We have now added the raw ITC data in the supplementary figure S7.

Reviewer #3 (Remarks to the Author):

In this work, Rustemi et al. employed the PRISMA method to study the influence of mutations and phosphorylation on protein-protein interactions. The authors started from a mutation database to select 38 peptides with unique mutations in intrinsically disordered regions and with specific phosphorylation, then prepared 117 peptide spots from the 38 peptides as well as one positive control EGFR-derived phosphopeptide, in wild type, with mutation and with phosphorylation to perform the PRISMA assay using SILAC for quantification. 170 differential specific interactions were identified using the HEK293 cell. Afterwards, the authors pay specific attention to the interaction between the 14-3-3 family of proteins and a GATAD1-derived phosphopeptide. It was found that the binding is regulated by the phosphorylation on GATAD1 S102, a modification site specifically observed in the heart in vivo. Finally, the authors show that the 14-3-3 binding region of GATAD1 is a functional nuclear localization signal, suggesting that binding of 14-3-3 proteins to phosphorylated GATAD1 could affect its subcellular localization. Overall, this is a very comprehensive work demonstrating the importance of phosphorylation and mutation in protein-protein interaction. The work also demonstrated the value of the PRISMA method in the study of protein-protein interactions. As pointed out by the authors, a main limitation of the current work is the lack of efficient in vivo model with the GATAD1 pS102. However, the authors performed sufficient biological experiments in vitro and in vivo to demonstrate their conclusion. The work is well designed, with comprehensive data, and the manuscript is well written. I have only a few comments for the authors' consideration.

We thank the reviewer for the encouraging comments!

1. Two quantitative filters were applied to identify proteins that exhibit both specific interactions with a particular peptide and are influenced by its phosphorylation and mutation state. Although it is reported in the previous publication, it can be briefly explained here to facilitate understanding the work.

Our quantitative filtering strategy is based on the fact that we have two different types of quantitative information: The differences in the abundance of proteins pulled-down with the different forms of the peptide (that is, wild-type, mutated and phosphorylated wild-type) and differences between completely different peptide candidates in the screen. For example, it would be possible that a protein binds more to the mutated form of a peptide than to the wild-type one. At the same time, this protein might also bind to many other peptide candidates from the screen, which would indicate that it does not bind specifically.

The two filters we applied were designed to avoid this problem. First, we used label free quantification to identify proteins that bind specifically to one peptide candidate. Second, we used SILAC to identify differential binders across the states (wt, mut, phos). For the interaction network we only considered proteins that are both specific (passing filter #1) and differential (passing filter #2). We already successfully used the same strategy in a previous study (Meyer et al., Cell, 2018).

Detailed information on the data analysis filters can be found in the section Data analysis of Material and Methods. To make this more clear in the main text we have now included the following sentences under the 'Quantification enables detection of specific interactions' section

“To achieve this, we utilized a Wilcoxon test to contrast the proteins derived from a single peptide with the background.”,

“Differential proteins were identified based on their log₂ wt/mut, wt/phos, and phos/mut ratios being either greater than 1, or less than -1.”, and

“More detailed information on the data analysis pipeline can be found in section 'Data Analysis' of Material and Methods.”

2. The BioID experiment was performed to study the interaction of 14-3-3 family proteins with GATAD1 phosphorylated on serine, and negative results were obtained due to the lack of pS102 in the HEK293 cell. However, the experiments may support the other findings in the PRISMA assay. Do you observe the interactions between the GATAD1 and other proteins by BioID as those by PRISMA? Some results can be presented.

We fully agree with this point, we also thought it would be interesting to compare the common interactors among all the methods, and that is why we had done some analysis in this direction. Below, we present a figure highlighting common interactors across these methodologies. In Panel A, we showcase the initial results of the PRISMA screen, including both Volcano plots and the SILAC ratio plot.

While the phosphorylated GATAD1 peptide uniquely interacted with the 14-3-3 proteins, the mutated GATAD1 peptide displayed no specific interactors, and wild-type GATAD1 peptide interacted specifically with five proteins, one of them being KNOP1. It's worth mentioning that while KNOP1 wasn't deemed significant in the PRISMA screen for the mutated peptide, its SILAC ratio WT/MUT was inconclusive. Hence, it is not a “differential” binder according to our stringent criteria (see answer to point 1 above), which is why it is not presented in the peptide-protein network.

This same protein, KNOP1, emerged as a significant interactor in the BioID experiment (evident from both 'no biotin' and 'no tetracycline' controls) as shown in the BioID scatter plot (bottom right). It also appeared in the alanine scanning but didn't qualify as significant, post ANOVA.

In our comparative analysis between PRISMA and Alanine Scanning, besides 14-3-3 proteins, MTX2 and RSL1D1 stood out (Panel B, right). While both MTX2 and RSL1D1 displayed significant SILAC ratios in the PRISMA screen, they did not exhibit specificity (LFQ filtering in PRISMA) to any peptide forms (that is, they are not specific according to our LFQ filter, see answer to point 1 above), as depicted in the volcano plots (Panel A).

And finally when comparing Alanine Scanning and BioID, we found commonalities in four proteins: MKI67, RUVBL1, PSME3, KHSRP, and NUMA1. Enrichment analysis of these candidates spotlighted 'regulation of cell cycle processes' as a prominent biological process annotation, with MKI67, PSME3, and NUMA1 clustering within.

While all these proteins are intriguing considering they could be potential novel GATAD1 interactors, we couldn't find solid evidence to connect them directly to GATAD1's function. Therefore, they were not extensively discussed in the original manuscript.

Figure R3: Shared proteins identified across PRISMA, alanine scanning, and BioID techniques.

A) Visual representations from the PRISMA screen are shown. The initial three graphs display the volcano plots derived from LFQ filtering. The final scatter plot illustrates the SILAC ratios, highlighting our second filtering strategy.

B) The scatter plot on the left displays the data from the PRISMA screen (identical to the one in panel A). Red dots indicate proteins that were deemed significant in the alanine scanning experiment, orange labels pinpoint proteins significant in the alanine scanning with notable SILAC ratios in the PRISMA screen. On the right, the scatter plot exhibits results from the BioID experiment. Orange labels here denote proteins significant in the alanine scanning, while blue labels mark those significant in the PRISMA screen.

3. From the 170 differential specific interactions, why a focus was given to the interaction between the 14-3-3 family of proteins and the GATAD1-derived phosphopeptide? It should be better explained.

This question is very relevant especially since when conducting such screens, distinguishing biologically relevant findings from noise and deciding on the next steps is always challenging. We actually looked at many interactions we identified and tried to build hypotheses how they could be linked to disease.

We actually mentioned a number of reasons why we focussed on 14-3-3 proteins and GATAD1 in the main text (first paragraph of the section “A phosphorylation site in the transcription factor GATAD1 interacts with 14-3-3 proteins”): We now extended this section as follows:

“To further investigate the potential function of novel interactions, we directed our attention to the binding of multiple members of the 14-3-3 family of proteins to a GATAD1-derived phosphopeptide (Figure 4A and S3). 14-3-3 family proteins are important regulatory molecules involved in a staggering number of cellular processes that interact with target proteins in a phosphorylation-dependent manner (Pennington et al. 2018; Ballone, Centorrino, and Ottmann 2018; Fu, Subramanian, and Masters 2000). GATAD1 is a transcription factor affecting proliferation and cell cycle via controlling AKT signaling (Sun et al. 2018). The GATAD1 S102P mutation we investigated in the screen was described to cause dilated cardiomyopathy (DCM) in an autosomal recessive manner in a consanguineous family (Theis et al. 2011). Interestingly, the mutation appears to affect the subcellular localisation of GATAD1 in cardiomyocytes of patients carrying this mutation. This is interesting since 14-3-3 proteins have been shown to regulate the subcellular localisation of their binding partners. Experiments in zebrafish provided additional evidence for the pathogenicity of this mutation (Yang et al. 2016). However, neither the pathogenic mechanism nor the function of this GATAD1 phosphorylation site are currently known. Intriguingly, although GATAD1 is expressed in many tissues, the only evidence for phosphorylation of this site comes from murine heart tissue, indicative of a heart-specific function (Lundby et al. 2013; Hornbeck et al. 2015).”

Reviewer #4 (Remarks to the Author):

In this very interesting work, Rrustemi et al. investigate the phospho-dependent peptide-protein interactions focusing on a subset of peptides harboring pathogenic mutations in intrinsically disordered regions. Through databases analyses and iterative selection based on the aforementioned criteria (plus no additional mutations flanking the phosphorylated residue), they selected disease candidate peptides to perform PRISMA analysis. Each peptide was produced on a membrane in 3 versions: wt, wt phosphorylated and mutant. Following a methodological validation on a well-known EGFR phosphosite, they ran their pipeline on the remaining 37 peptide triplets. Their approach is systematic and provides valuable information, as they identify 70/111 peptides showing at least one specific interactor and 31/37 triplets for which one variant has preferential partner(s). Expectedly and supporting their approach, bait peptides containing

SH2 domain binding motifs are enriched in partners containing SH2 domains, [S/T]P containing peptides capture PIN1 and they explain the frequent binding of MMTAG2, ARL6IP4 and PC4 to negatively charged (phosphorylated) peptides through their stretch of basic residues. The authors then decided to follow up on GATAD1 since its pS102 peptide interacts with several 14-3-3 proteins. They performed BioID on 4 different constructs: GATAD1 WT, S102A (phosphodead), S102D (phosphomimetic) and S102P (natural mutant). Unexpectedly, the WT and S102D mutant did not detect the 14-3-3 proteins as proximal interactors. The authors then performed ITC between the different GATAD1 peptides and YWHAE, which convincingly shows that the phosphomimetic mutant do not recapitulate the ability of the phosphopeptide to interact with GATAD1. Furthermore, through alanine (glycine for alanine) scanning, they identified additional residues (+1A and +2P) critical in the epsilon 14-3-3 interaction. The structural analysis revealed that none of the non-phosphorylated variant could form an interaction with YWHAE given the distance between the residue 102 and the positively charged triad patch of YWHA. To gain insight into the functional significance of the pS102 peptide-14-3-3 interactions, the authors finally performed a PRISMA analysis with all the alanine scanning peptides + the variant pS102S/A/D/P, which they prove were unable to bind the 14-3-3 proteins. Strikingly, they identified an enrichment in nuclear import proteins suggesting that the binding of 14-3-3 proteins on the pS102 peptides could impair the nuclear import of GATAD1. Using the NLStradamus model, they identified a region spanning the 14-3-3 interacting peptide as a candidate NLS, which they confirming as a bona fide NLS fusing it to GFP. They finally conclude that the 14-3-3 could mask the pS102 GATAD1 NLS, impairing its proper transport into the nucleus.

It is notoriously difficult to study PTM-dependent interactions. I find the approach of the authors interesting and I acknowledge their application to present both positive (PRISMA, alanine scanning and structural analysis) and negative data (BioID). Their work appears conceptually important to me. For example, the ITC experiments are clearly showing that phosphomimetic peptides are not recapitulating all the features of a true phosphorylation, structurally well supported by the lack of proximity between the negative charge and the modified residue. Overall, I believe this work is of high quality, despite a few limitations that the authors are aware of (e.g. lack of model system) and a few reserves listed below that I think should be lift to reach the publication level of Nature Communications and make this paper more impactful.

In summary, the pathophysiological mechanism of this mutation could come from the defect of cytoplasmic sequestration of GATAD1 in cardiomyocytes, from the expression of a S102P variant in the nucleus or from the absence of the GATAD1wt. There is no insight about what mechanism is at play. However, even if the biological conclusion of this paper falls short and that interactomic study needs to be reworked, the methodological approaches and the results are of high quality. The follow up on GATAD1-14-3-3 interaction is complicated and the authors proposed a convincing mechanism that could be supported by a few additional assays. I would thus consider this paper suitable for Nature Communications following the revisions listed below.

We thank the reviewer for his/her supportive comments!

Major comments:

The BioID part of the manuscript is puzzling me. I find it laudable that the authors tried to investigate the relevance of the GATAD1 peptide variants using whole protein variants expressed in living cells and that they include these data to the manuscript, even if they are not really supportive of their PRISMA findings. Given the results obtained in the rest of the manuscript, the data are not very surprising. However, I have been through the BioID supplemental table, which contains very few information, then looked at the proteingroup file deposited on the proteomeXchange server.

- I have been trying to understand where did the ratio presented in the supplemental table come from and was unable to find the way that the authors used to get these results. Please include additional tabs with the data used to calculate these ratios and explain how you calculated them.

We apparently failed to make this clear enough. We acknowledge that the supplementary files are not intuitive, especially since we included only the processed data. We now made this clearer, included the “raw” ratios and added a “read me” legend at the beginning of the table.

The BioID data are now in Supplemental table 7. The ProteinGroups sheet is the MaxQuant output table “proteinGroups.txt”. This is the data table that we worked with. This table contains 3053 proteins. We filtered this table for potential contaminants, only those identified by site and those identified from the reverse database. This reduced the number of proteins to 2809. From this table, we then extracted the normalized SILAC ratios as provided by MaxQuant. The columns named “Ratio.H.L.normalized.WT_...” contain these SILAC ratios. We had 5 conditions/comparisons: WT vs ‘no biotin’ control, WT vs ‘no tetracycline’ control, WT vs S102P, WT vs S102D and WT vs S102A. Since we did the label swap, there were two replicates for each comparison named “forward” and “reverse”. These SILAC ratios are then log₂ transformed, and the log₂ values of the forward experiments were multiplied with -1 (since in the forward experiment the light state was the wild-type and the heavy state the mutant or control). Both SILAC ratios and the log₂ transformed ratios can be found in the ‘SILACRatios’ sheet of the Supplemental table 7. The scatter plots in Figure S6 depict log₂ SILAC ratios from the forward (x-axis) and reverse (y-axis) experiment for each condition. Proteins with a Log₂ ratio >1 or < (-1) are considered as differential. These proteins are extracted and reported for easier visualization in the other workbook sheets of the supplemental table 7. For WT vs ‘no biotin’ control the number of proteins surpassing this cut-off is 135.

- The common way of doing BioID is filtering out the background using the BirA* tag alone, not the ‘no biotin’. I do not understand how your data could define the basal proximal interactome of GATAD1 comparing GATAD1wt to ‘no biotin’.

This point is well taken, especially since selecting the “right” control is a critical factor in any BioID experiment. However, we think it is important to point out that this choice is context-dependent, so there is no universally applicable “right” control. There are a number of points that we think are relevant in this context.

1. To study the interactome of GATAD1 variants we have generated stable polyclonal cell lines that express these protein variants. All these cell lines differ from one another to some extent. Using a 'no biotin' control is advantageous and more specific because it allows us to compare the same stable cell line, WT 'with' and WT 'without biotin' addition. Using other controls would be affected by differences between cell lines.
2. Additionally, the cytosolic BirA might not be the best control for a nuclear protein, as a nuclear protein would not be the best control for a cytosolic protein. In reality, many proteins (including GATAD1) can be found both in the cytosol and in the nucleus, which further complicates the choice of a universal control. Arguably, the best control for the impact of a mutation on the subcellular localisation of a protein is to compare the wt and the mutated BioID constructs with each other, which is exactly what we did.
3. Having said this, we also have a 'no tetracycline' control (no induction of the BirA-fusion protein) which shows similar results to our 'no biotin' control. We have now added the data of this control too, both in the supplementary figure S6 and Table S7. The overlap between the 'no biotin' and 'no tetracycline' controls is very high, with 106 common proteins identified as differential for WT GATAD1 in both (Figure S6, pie chart).
4. Additionally, in our scatter plots we apply predefined cut-offs and we focus only on those proteins that pass this threshold. To obtain a more global view of the data without using cut-offs we also projected our SILAC ratios into the t-SNE plot generated by Go et al., as part of the "Human Cell Map" project (Go et al., Nature, 2021). This projection allowed us to determine the cellular distribution of the proteins identified with our BioID experiment and observe the enriched structures. The generated t-SNE plots are shown below. In these plots, it can be observed that both 'no biotin' and 'no tetracycline' controls behave similarly, and the proteins observed as differential for WT belong to 'chromosome,' 'nuclear body,' 'nucleolus,' and 'nucleoplasm' cellular structures. This aligns well with GATAD1 proteins localization and function. We could also observe that S102A GATAD1 variant behaved very similarly to the WT GATAD1.

Figure R4. Mapping of the BioID results onto the tSNE proximity plot (Go et al. 2021). The top figure displays the original tSNE, highlighting the various compartments. The three plots beneath showcase the projected BioID ratios for the specified conditions. Compartments that differ between WT and Control are emphasized. It's evident that S102A mirrors the behavior of WT GATAD1.

-I do not understand how you pass from 3,052 protein groups identified across the conditions to 183 (actually 152 because 31 are NA for all baits according to the supplemental table) protein groups in the table using the 'no biotin' condition as a sole filtering condition.

We now provide more detail in the supplemental table, which should make it easier to follow how we arrived at our numbers (see answer above). In short, we filter the ProteinGroups data table for potential contaminants, only identified by site, and identified by the reverse database, then we extract the SILAC ratios from the same ProteinGroups data table. We logarithmized these ratios and we swap the label of the forward experiment by multiplying the values with (-1) so that it reflects the ratios from the reverse experiment. Finally we select only those proteins that contain a logarithmized SILAC ratio bigger that 1 (differential for WT) or smaller than (-1) (differential for 'no biotin' control). This selection led to 135 proteins being differential for WT and 0 for 'no biotin' control.

In the previously provided supplementary table, 183 is the total number of the significant proteins among all the conditions. The NAs (48 NAs) in the 'no biotin' control column mean that these particular proteins are not significant for WT vs no biotin control, but are significant for some other condition, leaving 135 proteins significant for WT vs no biotin control. The same is true for other conditions.

We are not sure which 31 NAs for all the baits the reviewer is referring to. In any case, the confusion hopefully will be solved with the newly provided supplemental table S7.

- Performing SILAC for BioID is not common, but I acknowledge that it could help multiplexing and maybe reducing the MS analysis cost. This is at the expense of analysis depth, but it is not the main issue in interactomics studies.

The main reason we opted for SILAC-based quantification is that it actually is more accurate and precise than label free quantification. In particular, SILAC is great for quantification of small changes in protein levels between two forms such as wt and mutated form. Sample are combined at an early experimental stage, which removes any possible variability that could be caused by non-identical sample handling. The unique advantages of SILAC for functional and quantitative proteomics are discussed in this review paper that we also cited:

<https://www.nature.com/articles/nrm2067>

Two previous studies highlight the unique strengths of SILAC-based quantification for BioID (Meyer et al., Cell, 2018, <https://doi.org/10.1016/j.cell.2018.08.019>; Vieira et al., Mol Cell., 2022, <https://doi.org/10.1016/j.molcel.2022.03.024>). In these studies, we used the same SILAC-BioID approach to assess how point mutations affect protein neighborhood. The latter paper actually used a phosphomimetic mutation, and in this case it did capture relevant aspects of phosphorylation and resulted in a changed BioID signal. This further validates the approach we took. We show the corresponding figures from these previous papers below.

[REDACTED]	[REDACTED]
Figure 4C from Meyer et al. (Cell, 2018). This figure	Figure 4E from Vieira-Vieira et al. (Mol. Cell, 2022). This

shows how SILAC BioID identifies differential interactors of the wild-type and the mutated form of the glucose transporter GLUT1.

figure shows how SILAC BioID identifies differential interactors of wildtype RBM20 and a pathogenic variant (S635A), a non-phosphorylatable variant (A10) and a phosphomimetic variant (D10). SILAC log2 fold changes are plotted on top of the Cell Map data (see above).

R5 Two previous studies showing that combining BioID with SILAC-based quantification can reveal the impact of point mutations on protein neighborhoods.

- The only valuable data of the BioID part are the comparison between the mutants and the WT, but again, I would need additional information to understand where the ratios came from.

We do agree that the most valuable aspect of the BioID data is the comparison between the mutants – this was the main reason for performing these experiments. We hope that the additional explanations we provided make it easier to understand where the ratios come from. Having said this, we actually do see known interaction partners of GATAD1, which shows that also the “basal proximal interactome” of GATAD1 is valuable. But this is indeed not a central aspect of our paper.

- From my understanding, the label swap means that in one sample, WT is heavy and the other condition is light isotope, and in the other, the WT is light and the other condition is heavy. I was not able to identify what column corresponded to what condition.

This is correct. It is now explained in the “read me” section of supplemental table 7 which experiments are forward and which are reverse. Information on the SILAC labels for each experiment are also included in the “read me” section of the table.

Overall, I have been struggling a lot to get a feel about these data. Although from the presented results, it seems that the variants do not change much the interactome (which sometimes happens), I am unable to validate the results in their current form. At the very least, the authors should present their data in a comprehensive manner, using additional tabs to show how they arrived to their hit list from the MaxQuant output (there are no statistical values either). I have been sincerely trying to find the authors’ way using the MaxQuant output but unsuccessfully.

We apologize for not making this clear in the original submission. We hope that the additional details we now provided clarify this point (also see above).

In addition, they could either remove the part on the basal interactome of GATAD1 WT or do it as suggested below, because comparing the data between the bait and ‘no biotin’ is not the right way to do it. The comparison between the WT and mutants is what really matters here, and even if there are only duplicates (acceptable for MS data), the important result is that they see no real differences (I would like to see the calculation process to get to this conclusion though).

As stated above, we hope that the additional information we provide makes it more transparent how exactly we analyzed the data. We would also like to point out that due to the higher precision and accuracy, SILAC-based quantification requires fewer replicates than label free quantification. We are mentioning this because it appears that this reviewer has more experience with label free quantification, which is intrinsically less precise and accurate and therefore requires more replicates to compensate for this.

If they choose to do these corrections and not perform additional BioID experiments, I would suggest putting this part of the results in supplemental data and presenting it as a pilot experiment (since the main result is that the mutants do not follow the PRISMA interaction pattern with the 14-3-3 proteins, and that they further show that it is due to (i) an imperfect mimicking of phosphorylation by the S102D; (ii) a transport of all constructs in the nucleus).

To make the experiments done in cells insightful, I would suggest two different options:

1. If the authors really want to keep the BioID data in the core of their paper, I think they could quickly fix the aforementioned issues since they have made the stable cell lines. They can redo the whole BioID part, using regular culture conditions (not SILAC) and analyze independent triplicates for each cell line, adding a triplicate of BirA* alone samples as controls (that would be 15 samples). Alternatively, for the controls, it would also be acceptable to use files from the CRAPome depository (Mellacheruvu et al., Nat Methods, 2013; <https://reprint-apms.org/>) to remove background (there are BirA* alone files in HEK293 Flp-In TReX cells in there, but make sure they have been analyzed on an Exploris). The comparison of BirA*-GATAD1 with BirA* alone would provide the real basal proximal interactome of GATAD1. The 'no biotin' or 'no tet' controls are not necessary since the BirA* alone samples will be more stringent. Just use 50µM biotin everywhere.

2. If they opt for moving the BioID data in supplemental, I think they should show additional supportive evidence that the GATAD1 NLS they propose is functional, e.g. through transiently transfecting a candidate NLS mutant into HEK293 cells along with the other variants they used for BioID. Also, given that the S102E variant was structurally better at mimicking the phosphosite, I would add it in this experiment (or ideally as an additional BioID bait).

Suggestions: This is a little beyond the main message of the paper, so, unless the authors have already done some of the following experiments (even if the data are negative), I do not expect them to be included in a revised version. All the experiments in cellular systems could be performed in GATAD1wt depleted cells. According to Theis et al., the S102P mutation is pathogenic when homozygote in patients. This suggests that performing a set of experiments in a GATAD1 KD or KO background could be the best way to gather biologically relevant data in cellular systems. BioID could be more efficient using the TurboID tag (Branon et al., Nat Biotech, 2018; ~30 min labeling time instead of 18 h for regular BioID), in case the residence time of GATAD1 is too short for sufficient labeling in the cytoplasm. In these settings, it might be worth performing the analysis on cytoplasmic and nuclear fractions.

Finally, there is the common issue of in vitro binding analyses between proteins that are localized in different subcellular compartments. Biochemically possible interaction could be biologically rare or impossible. I think this is the case for GATAD1 (nuclear) and the 14-3-3 proteins (cytoplasmic). As the authors state, the model system is lacking and they cannot easily investigate this interaction in living cells. It could indeed be specific of the cardiomyocytes since this mutation is linked to dilated cardiomyopathy, but the biological evidences are lacking. The hypothesis that the 14-3-3 proteins could mask the NLS of GATAD1 suggests that a cardiomyocyte-specific kinase could phosphorylate this residue. From the PRISMA data, the authors' choice is obvious but the discrepancy between the basal localization of the 14-3-3 proteins and GATAD1 render the functional studies much more complicated. This is why one of the two options proposed above appear necessary to support their results.

As outlined above, we do believe that our BioID is meaningful as is, and we do not think that a BirA* alone control would significantly improve it. Therefore, we decided not to perform additional BioID experiments. Nevertheless, we followed the suggestion of the reviewer and moved the BioID to the supplement. Instead, we now performed additional AP-MS experiments to study the impact of the different point mutations on GATAD1 interaction partners (new figure 4B). The AP-MS results we obtained are overall very similar to the BioID data, which further confirms the findings.

We also followed the second suggestion and performed additional experiments to further validate the function of the putative GATAD1 NLS. We had already shown that fusing this NLS sequence to GFP results in nuclear localisation. We now also performed a loss of function experiment (new figure 6 D): We deleted the putative NLS from full length GATAD1 and investigated the impact of this deletion on the subcellular localisation of the protein. We see that deleting the NLS results in relocalisation of GATAD1 to the cytosol. Hence, both experiments combined demonstrate that the putative NLS is both required and sufficient for GATAD1 localisation to the nucleus.

We also included a more detailed analysis of GATAD1 mutations on the subcellular localisation of the protein (new figure S5). These data show that all GATAD1 point mutations localise to the nucleus, similar to the wild-type protein. We can therefore conclude that the point mutations themselves do not appear to interfere significantly with nuclear cytosolic trafficking.

Finally, we also considered the suggestion related to the E mutant: The reviewer correctly points out that it is structurally more similar to phosphorylated GATAD1. However, our ITC data already showed that it does not bind to 14-3-3 proteins. We nevertheless tested if the E mutant might alter subcellular localisation of GATAD1. To this end, we transiently transfected the protein into HEK293 cells (see figure R6 below). We found that the protein is nuclear as all other GATAD1 point mutations we investigated. Therefore, we decided not to perform BioID experiments with the S102E mutant.

Figure R6: Subcellular localisation of FLAG-tagged S102 after transient transfection into HEK293 cells.

Minor comments:

- Page 32: 50 μ M biotin instead of 50mM

Thanks for spotting this – we corrected this!

- The streptavidin-sepharose beads from GE vary from lot to lot, sometimes they are coated with too many streptavidin which generates too many streptavidin peptides following on beads digestion (Saint-Germain et al., J. Proteome Res., 2020). I checked the MS raw files and they are fine. For future studies, I would recommend to use the Pierce Ultralink beads instead.

Thanks for the suggestion – we will consider this in future experiments!

- The Tomioka et al., reference is incomplete (BioRxiv; use the citation tool to import the ref)

Thanks again for spotting this. We fixed the broken reference.

- For the PRISMA analyses, I did not find the information of the HEK293 lysis buffer and how the amounts of protein from each labeled sample (H, M, L) were normalized.

This information can be found in the Material and Methods part (section “Experimental Setup”)

Cell lysis buffer: HEPES (50 mM, pH 7.9), NaCl (150 mM), EGTA (1 mM), MgCl₂ (1 mM), glycerol (20%), NP-40 (1%), SDS (0.1%), and sodium deoxycholate (0.5%).

The amount of protein in the cell lysate was measured with DC protein assay and a lysate of 8 mg/ml was used per membrane.

We added the following sentence in the material and methods to make this more clear

“HEK-293 cells were lysed with the lysis buffer and the protein concentration was measured using DC protein assay (Bio-Rad)”

- The supplemental table show minimal information. It would be better to put the raw output of MaxQuant, then an intermediary table showing the columns useful for calculations, then the hit table. This is pretty much impossible in the current form to understand for a non-MS specialist.

We have modified all the supplementary tables, and have added more information to make the content clearer. Additionally, we have also added a read me tab at the beginning of each table.

- Page 12, it is stated that “All of these peptides contain an [pS/T]P motif known to interact with the WW domain of PIN1 (Lee et al, 2011)”. However, in Figure 3C, 2 peptides do not interact with this motif. If correct, change to “Eight out of ten of these peptides...”

This is correct. The devil is in the details, as always: In total, there are actually 12 peptides interacting with PIN1 (see the network in panel A, which shows 12 edges for PIN1). 10 of these peptides contain a PIN1 motif (“WW->DOC_WW_Pin1_4”) and can therefore be explained by SLiM-domain pairs. These 10 are shown in panel 3C. While all 10 contain the PIN1 motif, 2 of them do not have the motif at the phosphorylation site. This is why we did not highlight the serine residues for these two peptides in panel 3C.

To clarify this, we changed the text as follows:

“10 of these peptides contain an [pS/T]P motif, known to interact with the WW domain of PIN1 (T. H. Lee et al. 2011). 8 out of 10 peptides contain the motif at the phosphorylated site, while 2 others contain it in the adjacent sequence.”

- Page 12: “MMTAG2, ARL6IP4 and PC4 interact with many phosphopeptides (9, 7 and 6, respectively).” I count 5 phosphopeptides for PC4. Maybe 2 edges are superimposed?

We have fixed the figure. It was a mistake that happened during the layouting in illustrator.

- Unless I missed them, please add the supplemental tables legends under the form of a “readme” tab.

We have added a “readme” tab in each of the supplementary tables.

- Why did you choose the YWHAE and not another 14-3-3 protein?

As mentioned in the manuscript, 14-3-3 proteins usually have similar specificity towards their targets. Therefore, we believe that whichever of the 14-3-3 proteins observed in the PRISMA screen we would have chosen, the results would have been similar. However, when making the decision we opted for YWHAE because it was the most abundant 14-3-3 identified in the screen and it is one of the most abundant in heart tissue (Thompson and Goldspink 2022).

REVIEWERS' COMMENTS

Reviewer #1 (Remarks to the Author):

Thanks for the nicely revised manuscript.

Reviewer #2 (Remarks to the Author):

The authors responded adequately to all my comments. I appreciate that they attempted to demonstrate the interaction of full-length GATAD1 with 14-3-3 in vitro. However, the newly added co-immunoprecipitation experiments combined with MS showed no differences between GATAD1 wt and the S102A mutant. The authors' proposed explanation that GATAD1 is not phosphorylated at Ser102 in HEK cells might be relevant, but the authors did not provide direct evidence that this is the case (e.g. by MS analysis of the co-immunoprecipitated protein). In any case, the possible interaction of GATAD1 and 14-3-3 through the phosphorylated Ser102 motif and the effect of this interaction on the cellular localization of GATAD1 may be answered by the following studies.

Reviewer #3 (Remarks to the Author):

The authors have addressed all my previous concerns by this revision. I have no further comments.

Reviewer #4 (Remarks to the Author):

I thank the authors for their detailed responses and explanations. The manuscript looks much clearer and impactful to me. All the questions I asked have been thoroughly solved. The authors added a comprehensive description of their data and meaningful additional experiments supporting their conclusions. In my opinion, this manuscript should be accepted in Nature Communications in its current form.